# COSMOBENCH: A Multiscale, Multiview, Multitask Cosmology Benchmark for Geometric Deep Learning

Ningyuan (Teresa) Huang[1]    Richard Stiskalek[2,3]    Jun-Young Lee[2,4,5]
Adrian E. Bayer[2,5]    Charles C. Margossian[1]    Christian Kragh Jespersen[5]    Lucia A. Perez[2,5]
Lawrence K. Saul[1]    Francisco Villaescusa-Navarro[2,5]

[1]Centers for Computational Mathematics, [2]Computational Astrophysics, Flatiron Institute
[3]University of Oxford    [4]Seoul National University    [5]Princeton University

## Abstract

Cosmological simulations provide a wealth of data in the form of point clouds and directed trees. A crucial goal is to extract insights from this data that shed light on the nature and composition of the Universe. In this paper we introduce COSMOBENCH, a benchmark dataset curated from state-of-the-art cosmological simulations whose runs required more than 41 million core-hours and generated over two petabytes of data. COSMOBENCH is the largest dataset of its kind: it contains 34 thousand point clouds from simulations of dark matter halos and galaxies at three different length scales, as well as 25 thousand directed trees that record the formation history of halos on two different time scales. The data in COSMOBENCH can be used for multiple tasks—to predict cosmological parameters from point clouds and merger trees, to predict the velocities of individual halos and galaxies from their collective positions, and to reconstruct merger trees on finer time scales from those on coarser time scales. We provide multiple baselines on these tasks, some based on established approaches from cosmological modeling and others rooted in machine learning. For the latter, we study different approaches—from simple linear models that are minimally constrained by symmetries to much larger and more computationally-demanding models in deep learning, such as graph neural networks. We find that least-squares fits with a handful of invariant features sometimes outperform deep architectures with many more parameters and far longer training times. Still there remains tremendous potential to improve these baselines by combining machine learning and cosmological modeling in a more principled way, one that fully exploits the structure in the data. COSMOBENCH sets the stage for bridging cosmology and geometric deep learning at scale. We invite the community to push the frontier of scientific discovery by engaging with this challenging, high-impact dataset. The data and code are available at this URL.

## 1 Introduction

Cosmological simulations are powerful tools to model the distribution of dark matter and galaxies in the Universe. Astrophysicists use them as virtual laboratories to test hypotheses about the laws and constituents of our Universe—such as cosmic expansion and galaxy formation—which cannot be answered through observations alone. These simulations provide a wealth of data in the forms of point clouds and merger trees. Such data is ripe to be analyzed by methods in geometric deep learning [1], but cosmologists have yet to observe the same breakthroughs that these methods have produced in areas such as computer vision [2], structural biology [3, 4], and climate science [5]. In these areas, rapid progress has been driven by large-scale benchmarks that provide a unified interface to study multiple related tasks using machine learning (ML). Cosmology stands to benefit similarly,

39th Conference on Neural Information Processing Systems (NeurIPS 2025) Track on Datasets and Benchmarks.

as unified benchmarks can facilitate the exchange of ideas between cosmology and ML, driving the development of novel methods and deeper insights into the Universe.

For these reasons, we introduce COSMOBENCH, a benchmark for geometric deep learning curated from state-of-the-art cosmological simulations. In total, these simulations required more than 41 million core-hours and generated over two petabytes of data. COSMOBENCH is currently the largest multiscale and multiview benchmark in cosmology: it contains 34 thousand point clouds across three different spatial scales, as well as 25 thousand directed trees on two different temporal scales. Table 1 provides an overview of all the datasets in COSMOBENCH, and Fig. 1 provides an illustration of the data generation process. We briefly describe each of these sources of data (see App. A.1 for a glossary of cosmology terms).

The point clouds in COSMOBENCH encode the spatial distribution of either dark matter halos or galaxies, and they are arranged into three datasets based on the simulation suites (`Quijote` [6], `CAMELS-SAM` [7], and `CAMELS` [8]) that were used to create them. Each point cloud is characterized (or labeled) by the particular values of cosmological and astrophysical parameters that were used to govern the evolution of matter in its simulation. These simulations are not deterministic, so that a variety of point clouds may be generated by simulations with the same underlying parameters but different randomly sampled initial conditions.

The directed trees in COSMOBENCH are available only within `CAMELS-SAM`, and they are collected in another dataset, which we call `CS-Trees`. Each tree in this dataset represents the formation history of a present-time dark matter halo. The merger trees are constructed from 100 simulation snapshots and typically contain on the order of tens of thousands of nodes.

The goal of COSMOBENCH is to unlock the potential of ML in cosmology. With this goal in mind, COSMOBENCH consolidates multiple tasks of cosmological interest for ML at scale. These tasks include the prediction of cosmological parameters from point clouds and merger trees (graph-level regression), the prediction of the velocities of individual halos or galaxies from their collective positions (node regression), and the reconstruction of fine-scale merger trees from those on coarser time scales (graph super-resolution). If ML can solve these tasks, then cosmologists will be able to extract more information from simulations and observational data, and from this information they will learn more about the fundamental laws and constituents of our Universe.

A few examples underscore the value of these tasks. First, if cosmological parameters can be inferred from simulated point clouds and merger trees, then one could apply the same models to infer the parameters from observed galaxy distributions [9–11] or even the present-time properties of a single galaxy [12]. Second, if galaxy velocities can be predicted from galaxy positions, then this extra information can be used to vastly improve understanding of the structure and rate of expansion of the Universe [13–18]. Finally, if merger trees on finer time scales can be predicted from those on coarser scales, then one could compensate for the hardware constraints when producing simulations for upcoming cosmological surveys—for observatories such as *Euclid* [19] or *LSST* [20]. All of these are important problems in cosmological research.

We provide multiple baselines for all the tasks in COSMOBENCH. These baselines include well-established approaches in cosmology, simple linear models constrained by physical symmetries, and deep learning models such as graph neural networks. Notably, we find that least-squares fits with a handful of invariant features sometimes outperform graph neural networks with hundreds of thousands of parameters and significantly higher computational costs. Nonetheless, there remains huge potential to improve these baselines via more principled methods that combine ideas from ML and cosmology. We invite the community to push the frontier of cosmology by engaging with COSMOBENCH. Our key contributions are summarized as follows:

- We present COSMOBENCH, a multiscale, multiview, multitask benchmark for ML to push the frontiers of cosmology. The datasets are available at this URL.

- We provide a unified `PyTorch` interface to implement our proposed baselines on multiple tasks, including cosmological parameter prediction, halo/galaxy velocity prediction, and node classification in merger trees. This interface is available at this GitHub repository.

- On these tasks we show that ML-based methods sometimes outperform more established models in cosmology or require less information to solve the same problem. We also find evidence that simple linear models with invariant features excel at predictions on larger spatial scales, whereas graph neural networks are more effective at smaller spatial scales.

Table 1: Summary of Datasets in CosmoBench

| Dataset | Quijote | CAMELS-SAM | CAMELS | CS-Trees |
|---|---|---|---|---|
| Modality | Point Clouds | | | Directed Trees |
| Box Size | 1,000 cMpc/$h$ | 100 cMpc/$h$ | 25 cMpc/$h$ | 100 cMpc/$h$ |
| Node Entity | Halo | Galaxy | Galaxy | Halo |
| Number of Graphs | 32,752 | 1,000 | 1,000 | 24,996 |
| Number of Nodes | 5,000 | 5,000 | [588, 4,511] | [121, 37,865] |

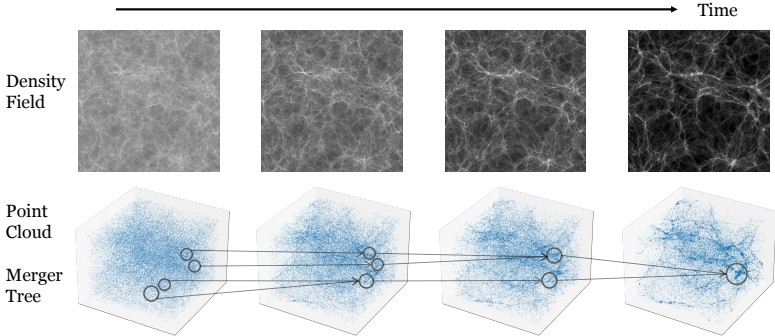

Figure 1: Illustration of point clouds and merger trees obtained from cosmological simulation.

## 2 Related Work

**Geometric Deep learning for Cosmology** Geometric deep learning has emerged as a promising tool in cosmology, since the cosmic web, along with the dark matter halos and galaxies embedded within it, can be more efficiently described as point clouds than regular grids. This has motivated extensive work in learning accurate mappings from point clouds representing galaxy distributions to cosmological parameters [21–28], accelerating and improving the necessary modeling of galaxies [29–34], optimizing observations of the large scale structure of the Universe [35, 36], or reconstructing strong lensing signals [37]. However, recent works also show that off-the-shelf methods from geometric deep learning sometimes struggle with various tasks, such as predicting cosmological parameters solely from point cloud positions [23, 38]. To systematically evaluate deep learning methods, we introduce baselines from cosmology and simple linear models built on interpretable, symmetry-constrained features.

**Benchmarks on Point Clouds** Popular benchmarks on point clouds mostly come from computer vision (e.g., ShapeNet [39], ModelNet [40]) and structural biology (e.g., AlphaFold DB [3], Molecu-leNet [41], Atom3D [42]). Despite the abundance of point clouds from cosmological simulations [6–8], there is no unified benchmark that formalizes cosmological tasks for ML at scale. A notable recent effort by Balla et al. [43] introduced a subset of 3,560 point clouds from Quijote [6] to evaluate equivariant ML models on predicting cosmological parameters and velocities of individual halos. Yet this benchmark is limited in size, physical scale, data modality, and task variety. Our benchmark significantly expands over Balla et al. [43] by providing (i) more than 34 thousand point clouds across three simulation suites that model the Universe from linear to deeply non-linear scales, (ii) halo merger tree data, and (iii) more diverse tasks and baselines.

**Benchmarks on Graphs** Point clouds and directed trees—the central objects in CosmoBench— are structured variants of graph data. While there are many large-scale graph benchmarks [44–47], most of them are limited to applications in biology and social science. Some graph benchmarks also suffer from improper graph construction or unreliable benchmarking protocol, as pointed out by [48]. CosmoBench offers a high-quality graph benchmark for challenging problems in cosmology, with varying graph structures, diverse tasks, and a wide spectrum of baselines. It responds to the call in [48] for improving graph benchmarks to spur further progress in graph ML.

# 3 Proposed Dataset and Tasks

## 3.1 Multiscale Point Cloud Dataset

We consider three cosmological simulation suites: `Quijote` [6], `CAMELS-SAM` [7], and `CAMELS` [8]. Each simulation is defined by a set of cosmological (and astrophysical in `CAMELS` and `CAMELS-SAM`) parameters and random initial conditions, which are then simulated up to present time to produce theoretical predictions for the temporal evolution and spatial distribution of halos or galaxies. The simulations differ in several aspects, including the number of sampled cosmological parameters (five in `Quijote`, two in `CAMELS` and `CAMELS-SAM`), the type of simulation ($N$-body simulations in `Quijote` and `CAMELS-SAM`, hydrodynamical in `CAMELS`), as well as box size, mass resolution, and time resolution. In particular, the time resolution (limited by storage constraints) affects the level of detail of merger trees. In what follows, we provide a high-level description of these simulation suites.

`Quijote`. The `Quijote` Big Sobol Sequence (BSQ) suite comprises 32,768 dark matter-only $N$-body simulations, each with a box size of $1000 \text{ cMpc}/h$ and dark matter particle mass of $\sim 10^{12} h^{-1} M_\odot$, depending on the value of $\Omega_{\mathrm{m}}$. The individual simulations are relatively inexpensive to run due to their low mass resolution, though the full suite required over 35 million core-hours. We exclude 16 simulations due to failures in the halo-finding algorithm, leaving 32,752 simulations in the final dataset. In total five cosmological parameters ($\Omega_{\mathrm{m}}, \sigma_8, \Omega_{\mathrm{b}}, h, n_s$) are sampled in `Quijote`.

`CAMELS-SAM`. The `CAMELS-SAM` suite consists of 1000 dark matter-only simulations with a box size of $100 \text{ cMpc}/h$, dark matter particle mass of $\sim 10^8 M_\odot/h$, and 100 stored snapshots per simulation for constructing dark matter halo merger trees. These trees serve as input to the `Santa-Cruz` semi-analytical galaxy formation model [49], which is used to generate synthetic galaxy populations. Compared to `Quijote`, the higher mass resolution of `CAMELS-SAM` allows it to resolve smaller, highly non-linear scales, at the expense of simulating a smaller volume. Only two cosmological parameters are varied in `CAMELS-SAM` dark matter simulations ($\Omega_{\mathrm{m}}$ and $\sigma_8$), though three astrophysical parameters are also varied within the semi-analytical galaxy formation model.

`CAMELS`. The `CAMELS` TNG suite comprises 1000 cosmological hydrodynamical simulations that model the evolution of dark matter, gas, stars, and black holes. To mitigate the high computational cost of hydrodynamical modeling, the simulations are performed in relatively small boxes of $25 \text{ cMpc}/h$. The suite varies two cosmological parameters ($\Omega_{\mathrm{m}}$ and $\sigma_8$) alongside four astrophysical parameters.

**Dataset Creation and Split** In COSMOBENCH, the `Quijote`, `CAMELS-SAM`, and `CAMELS` datasets contain 32,752, 1,000, and 1,000 point clouds, respectively, from the corresponding simulation suites. Each point cloud is derived from the present time ($z = 0$) simulation snapshots and labeled with its cosmological parameters, whereas the node features describe the 3D position and velocity of halos (or galaxies). For `Quijote` and `CAMELS-SAM` datasets, we provide two samples for computational tractability: the coarse resolution clouds which contain the 5,000 most massive halos, and the fine resolution ones store all halos with up to 112,000 points in `Quijote` and 19,000 points in `CAMELS-SAM`. Each dataset is randomly split into training/validation/test sets with a 60/20/20 ratio. The same split is used across coarse-grained and fine-grained samples.

**Task and Evaluation** Given the point cloud input with 3D positions $\{\mathbf{x}_i = (x_i, y_i, z_i) \in \mathbb{R}^3\}_{i=1}^n$ stored in rows of a matrix $\mathbf{X} \in \mathbb{R}^{n \times 3}$, we consider a graph-level task and a node-level task. The graph-level regression aims to predict the cosmological parameters $f : \mathbf{X} \mapsto \mathbf{y} = (\Omega_{\mathrm{m}}, \sigma_8)$. The parameter $\Omega_{\mathrm{m}}$ represents the fraction of the Universe's energy in the form of matter, and $\sigma_8$ measures the amplitude of matter density fluctuations. The node-level regression task aims to predict the 3D velocity for each point in the cloud, $f : \mathbf{X} \mapsto \mathbf{V} \in \mathbb{R}^{n \times 3}$, where the $i^{\mathrm{th}}$ row of $\mathbf{V}$ stores the velocity $\mathbf{v}_i$ of the $i^{\mathrm{th}}$ point. For each task, the model is evaluated on the same test set, using the coefficient of determination $R_y^2$ for the cosmology prediction and $R_{\mathbf{v}}^2$ for the velocity prediction, defined as

$$R_y^2 = 1 - \frac{\sum_{i=1}^{n_{\text{test}}} (f(\mathbf{X}_i) - \mathbf{y}_i)^2}{\sum_{i=1}^{n_{\text{test}}} (\bar{\mathbf{y}} - \mathbf{y}_i)^2}, \quad R_{\mathbf{v}}^2 = 1 - \frac{\sum_{i,j}^{n_{\text{test}},3} (f(\mathbf{X})_{ij} - \mathbf{V}_{ij})^2}{\sum_{i,j}^{n_{\text{test}},3} (\bar{\mathbf{V}}_j - \mathbf{V}_{ij})^2}. \tag{1}$$

To provide uncertainty estimates, we use the standard deviation (std) of the $R^2$ on bootstrap data drawn from the test set.

## 3.2 Merger Tree Dataset

**Data Creation and Split** We select all merger trees from `CAMELS-SAM` with the root node (i.e. the present-time halo at redshift $z = 0$) having mass larger than $10^{13}$ $M_\odot/h$ ($\sim 10^5$ dark matter particles) to ensure that the evolution of the root node's progenitors is sufficiently well resolved. This yields a collection of more than 460,000 trees across 1,000 cosmological parameter combinations. Since some parameter choices produce significantly more large trees or heavier roots, we randomly select 25 trees per simulation with cosmological parameters $\mathbf{y} = (\Omega_m, \sigma_8)$. The original merger trees have up to hundred thousands of nodes and consist of long paths. To reduce storage requirement and condense information, we trim the trees by first removing subtrees with all nodes having mass less than $10^{10}$ $M_\odot/h$. We then prune each path where nodes with mass less than $3 \times 10^{10}$ $M_\odot/h$ are removed. See Fig. A.1 for an illustration. This procedure reduces the tree size by a factor of 5-10 and prevent information leakage[1], while maintaining most of the merger nodes relevant for probing the merger history, replacing the smooth halo evolution across a path with a single edge. We then remove any trees with less than 100 nodes. The final `CS-Trees` dataset contains 24,996 trimmed trees, split per their cosmological parameter choices $(600/204/196)$ (same as the `CAMELS-SAM` point cloud dataset split).

**Task and Evaluation** For merger trees, we consider a graph-level regression task to predict the cosmological parameters from tree inputs. We also consider a node-level classification task to determine missing merger nodes, motivated by merger tree temporal "super resolution" [50, 51]. For this task, we select the 200 largest trees from `CS-Trees` to avoid degeneracy. Specifically, given a (binary) merger tree $\mathcal{T}_0$, we coarsen it by masking out all even time steps, effectively reducing the temporal resolution by half. We arrive at a new (connected) tree by connecting the kept ancestor nodes to their next kept nodes, and save the unique post-merger node for each masked merger node in a dictionary $D$. We then binarize the coarsen tree on nodes with in-degree larger than 2, by keeping the top two pre-merger nodes ranked by their masses, resulting in $\mathcal{T}$. Finally, we append virtual nodes to each merger nodes in tree $\mathcal{T}$, and obtain $\mathcal{T}_c = (V + \tilde{V}, E + \tilde{E}, X \cup \mathbf{0})$. The virtual nodes represent the potentially unresolved mergers. We label a virtual node $\tilde{v} \in \tilde{V}$ as positive if its post-merger node is in the dictionary $D$, and negative otherwise. See App. A.3 for details of the full procedure. The node classification task builds a binary classifier for the all virtual nodes given the coarsened (binary) tree. We split the 200 trees randomly into 120/40/40 training/validation/test sets.

## 4 Point Cloud Baselines and Results

### 4.1 Predicting Cosmological Parameters from Point Clouds

**Two-Point Correlation Function** The task of point cloud regression, particularly of inferring cosmological parameters from the clustering of objects, is traditionally approached via the two-point correlation function $\xi(r)$ (2PCF). This statistic quantifies the excess probability $dP$ of finding a pair of points separated by a 3D distance $r$, relative to a uniform random distribution [52, 53]. While a Gaussian field is fully characterized by its 2PCF, the late-time distribution of halos or galaxies is highly non-Gaussian, particularly at small scales. Consequently, 2PCF becomes an insufficient statistic for capturing the full information content of the field. To address this limitation, higher-order correlation functions [7, 10], halo and void abundances [54–56], other alternative clustering statistics [57, and references within], or field-level inference methods are typically employed [58–61]. Recently, ML has offered promising approaches for mapping point clouds directly to cosmological parameters, with the potential to capture higher-order information [38, 24]. To predict cosmological parameters from 2PCF, we fit a multi-layer perceptron (MLP) with tunable hyperparameters; see App. B.1.

**Linear Least Squares with Pairwise-Distance Statistics** Motivated by the 2PCF, we also use simple linear least squares (LLS) models to predict the cosmological parameters $\Omega_m$ and $\sigma_8$ from the statistics of pairwise distances. For these models, we consider the empirical distributions of pairwise squared distances between points that are closer than some cutoff radius $R_c$. In particular, for each point cloud, and for different cutoff radii $R_c$, we compute the means, standard deviations,

---

[1]The simulation starts with particles whose mass is a simple function of the cosmological parameters; a halo is tracked once 20 particles are clustered close enough. Thus the ML model can "cheat" by focusing on halos of the smallest mass to predict cosmological parameters, instead of leveraging the halo point cloud structure.

and $(\frac{1}{3}, \frac{2}{3})$-quantiles of these distributions. To predict $\Omega_{\mathrm{m}}$ and $\sigma_8$, we extract 48 features from each point cloud by computing these 4 statistics for 12 different cutoff radii, and for each model, we select these cutoff radii greedily from the predictive accuracy of their statistics on the validation data. Finally, we use least-squares fits with a bias term to estimate linear models over these features, and we clip their predictions to lie within the limiting values of $\Omega_{\mathrm{m}} \in [0.1, 0.5]$ and $\sigma_8 \in [0.6, 1]$.

**Graph Neural Networks** To capture higher-order clustering information in point clouds, graph neural networks (GNNs) have recently shown encouraging performance for cosmology prediction [62, 63]. A point cloud naturally induces an Euclidean graph by linking pairs of points within some cutoff radius $R_c$, also known as a radius graph. We use the point positions as the node features $\mathbf{X} \in \mathbb{R}^{n \times 3}$. To respect the underlying translation and reflection symmetries of the cosmological point cloud, we decorate each edge with 3 features: the normalized pairwise distance $d_{ij}/R_c$, and the two dot products $\langle \mathbf{x}_i, \mathbf{x}_j \rangle$, $\langle \mathbf{x}_i, (\mathbf{x}_i - \mathbf{x}_j) \rangle$. To go beyond pairwise operation, we identify edge neighbors within the same tetrahedron from a 3-dimensional Delaunay triangulation. We also extract E(3)-invariant features from neighboring node-node, node-edge, and edge-edge pairs using Euclidean and Hausdorff distances, denoted as $\mathrm{Inv}(\cdot, \cdot)$.

Given such (higher-order) graphs, we apply GNNs with message-passing among nodes $V$ and edges $E$. Specifically, the message passing neural network maintains node embeddings $\mathbf{h_x} \in \mathbb{R}^{|V| \times d_n}$ and edge embeddings $\mathbf{h_e} \in \mathbb{R}^{|E| \times d_e}$. These embeddings are updated at the $l^{\mathrm{th}}$ layer as $\mathbf{h}_z^{(l)} = \mathbf{m}_z^{(l-1)} + \mathbf{h}_z^{(l-1)}$ where $z$ denotes a node or an edge, with the following message function

$$\mathbf{m}_z^{(l)} = \begin{cases} \bigoplus_{y \in \mathcal{N}_0(z)} \psi_{nn}(\mathbf{h}_y^{(l)}, \mathrm{Inv}(\mathbf{x}_z, \mathbf{x}_y)) \otimes \bigoplus_{y \in \mathcal{N}_1(z)} \psi_{en}(\mathbf{h}_y^{(l)}, \mathrm{Inv}(\mathbf{x}_z, \mathbf{x}_y)), & \text{if } z \in V \\ \bigoplus_{y \in \mathcal{N}_0(z)} \psi_{ne}(\mathbf{h}_y^{(l)}, \mathrm{Inv}(\mathbf{x}_z, \mathbf{x}_y)) \otimes \bigoplus_{y \in \mathcal{N}_1(z)} \psi_{ee}(\mathbf{h}_y^{(l)}, \mathrm{Inv}(\mathbf{x}_z, \mathbf{x}_y)), & \text{if } z \in E. \end{cases} \quad (2)$$

Here, $\mathcal{N}_0(x)$ denotes the neighboring nodes, and $\mathcal{N}_1(x)$ the neighboring edges. We use $\bigoplus$ for intra-neighborhood aggregation function and $\otimes$ the inter-neighborhood aggregation function; both are permutation-invariant. Finally, $\psi_{y,z}$ denotes a non-linear update function determined by the types of $y, z$, and $\sigma$ is an activation function. Our message-passing design is inspired by [64], which supports extensibility to neural networks operating on combinatorial complex topologies—a direction beyond the scope of our benchmarking and reserved for future work. We also perform an ablation on GNNs by removing edge-edge message-passing (denoted as "w/o edgeMP"). Full details on the network architecture and experiments are deferred to App. B.1.

**Results and Analysis** Table 2 reports the baseline results for predicting $\Omega_{\mathrm{m}}$ and $\sigma_8$. All of the above methods perform well on the large-scale point clouds in `Quijote`. In the higher-resolution but smaller-volume `CAMELS-SAM` simulations, the methods perform worse at predicting $\Omega_{\mathrm{m}}$ but similarly or better at predicting $\sigma_8$. In `CAMELS`, which probes the most non-linear scales, the predictions of $\Omega_{\mathrm{m}}$ are comparable to those in `Quijote`, but predictions of $\sigma_8$ are severely degraded. This is consistent with the expectation that $\sigma_8$ primarily affects rare, high-mass halos, of which only a few are present in `CAMELS` due to its small volume. Notably, the lightweight LLS model using pairwise statistics achieves performance competitive with GNNs requiring orders of magnitude more parameters and compute time. We also find no significant difference when disabling edge-edge message-passing in GNNs. We remark that a significant body of work has been carried out using the `Quijote` simulations[2] [6] to quantify the sensitivity of different summary statistics to $\sigma_8$. However, many of these were derived using Fisher matrix calculations and therefore we cannot perform an apple-to-apple comparison. In App. B.1, we include further ablation studies of 2PCF, LLS, and GNN, as well as the performance of these baselines for predicting three other cosmological parameters in `Quijote` to illustrate broader applicability yet increasing difficulty in constraining these parameters.

Table 2: Point Cloud Cosmological Parameter Regression ($R^2 \pm$ 1std). Time unit: 1-GPU time.

| $\mathbf{R^2}\uparrow$ | Quijote | | | | CAMELS-SAM | | | | CAMELS | | | |
|---|---|---|---|---|---|---|---|---|---|---|---|---|
| | $\Omega_{\mathrm{m}}$ | $\sigma_8$ | Params. | Time | $\Omega_{\mathrm{m}}$ | $\sigma_8$ | Params. | Time | $\Omega_{\mathrm{m}}$ | $\sigma_8$ | Params. | Time |
| 2PCF | 0.85 ±0.004 | 0.84 ±0.004 | 11K | 2 min | 0.73 ±0.03 | 0.82 ±0.02 | 10K | 10 sec | 0.84 ±0.02 | 0.30 ±0.06 | 8K | 10 sec |
| LLS | 0.83 ±0.004 | 0.80 ±0.004 | 49 | 24 sec | 0.77 ±0.03 | 0.82 ±0.02 | 49 | 3 sec | 0.78 ±0.03 | 0.28 ±0.06 | 49 | 3 sec |
| GNN | 0.80 ±0.004 | 0.77±0.005 | 671K | 1 day | 0.75 ±0.03 | 0.83 ±0.02 | 1003K | 3 hr | 0.78 ±0.03 | 0.24 ±0.06 | 1166K | 2 hr |
| GNN (w/o edgeMP) | 0.80 ±0.004 | 0.79±0.005 | 128K | 1 day | 0.72 ±0.03 | 0.84 ±0.02 | 506K | 3 hr | 0.80 ±0.02 | 0.27 ±0.06 | 384K | 2 hr |

[2]https://quijote-simulations.readthedocs.io/en/latest/publications.html

## 4.2 Predicting Velocities from Positions

**Linear Theory** In cosmology, the velocity field can be predicted from the matter field using the linearized continuity equation [65]. In the linear regime (i.e., large-scale structure such as the `Quijote` point clouds), where density fluctuations are small, mass conservation implies that the divergence of the peculiar velocity field $v$ is proportional to the time derivative of the density contrast $\delta$, i.e. $\nabla \cdot v \propto \partial \delta(x, t)/\partial t$. The solution to this is

$$v(x_i) \propto \int d^3 x_j \frac{\delta(x_j)(x_j - x_i)}{|x_j - x_i|^3},$$ (3)

where the constant of proportionality is dependent on the cosmological parameters. To provide a cosmology "oracle" for the benchmark, we solve this equation assuming perfect knowledge of cosmological parameters by fitting for the proportionality constant. We solve for the velocity field in Fourier space; further details are provided in App. B.2. This solution quantifies the most optimistic possible result achievable using linear theory, whose descriptions begin to fail at small scales. This limitation motivates moving beyond linear theory, for example using nonlinear theory with Bayesian inference [66, 67]; here we explore methods rooted in ML.

**Linear Least Squares with Powers of Inverse Distances** Inspired by linear theory, we also use a simple linear model to predict velocities from inverse powers of pairwise distances. Given the positions $\{x_i = (x_i, y_i, z_i) \in \mathbb{R}^3\}_{i=1}^n$, the model predicts the velocity $v_i$ of the $i^{\text{th}}$ halo or galaxy as

$$v_i = \sum_{p=1}^P \sum_{k=1}^K w_{pk} \sum_{j=1}^n \frac{\sin \frac{2\pi k}{B}(x_i - x_j)}{d_B(x_i, x_j)^p},$$ (4)

where $w_{pk}$ denotes the linear model's weights and $d_B(x_i, x_j)$ computes the wrap-around (toroidal) distance between $x_i$ and $x_j$ in a box of size $B$ with periodic boundary conditions. We use the validation set to choose the hyper-parameters $P \in \{2, 3, 4\}$ and $K \in \{10, 15, 25\}$.

**Graph Neural Networks** To match linear theory velocity prediction on large scales and surpass it at small scales, we apply message-passing neural networks (MPNNs) [68], with a message-passing scheme motivated from linear-theory and exploiting local neighborhood structure. To this end, we construct radius graphs induced from the clouds with radius cutoff $R_c = 0.1B$, where $B$ is the box size. Each radius graph has node features as 3D point positions $\{x_i\}_{i=1}^n$, and edge features as (rescaled) 3D position wrap-around difference, i.e. $e_{ij} = \frac{\text{sign}(x_i - x_j)}{R_c}[d_B(x_i, x_j), d_B(y_i, y_j), d_B(z_i, z_j)] \in \mathbb{R}^3$. The linear-theory inspired MPNN maintains the node embedding $h \in \mathbb{R}^{n \times d}$, which is updated at the $l^{\text{th}}$ layer as

$$h_i^{(l)} = \frac{1}{|\mathcal{N}(i)|} \sum_{j \in \mathcal{N}(i)} \phi(h_j^{(l-1)}) \psi(e_{ij}), \quad h_i^{(0)} = x_i,$$ (5)

where $\phi, \psi$ are MLPs. Further details are reported in App. B.2.

**Results and Analysis** The baseline results for velocity prediction are reported in Tab. 3. All baselines perform relatively well at large scales (`Quijote`) and deteriorate at smaller scales (`CAMELS-SAM` and `CAMELS`). Due to the large number of points in each point cloud, we omit the bootstrap uncertainties on $R^2$, as they are negligible. In `Quijote`, a simple LLS model with only a few parameters outperforms a significantly larger and more computationally expensive GNN. Both LLS and GNN models surpass the linear theory oracle, likely by capturing non-linear corrections to the velocities. Notably, while linear theory requires knowledge of the cosmological parameters, ML methods do not and still outperform it. The GNN is more effective than LLS in `CAMELS-SAM`, while in `CAMELS`, linear theory oracle surpasses both. In Fig. 2 we show the true and predicted projected velocity field in a single `Quijote` point cloud. We provide more visualizations, ablation studies, and discussions in App. B.2.

### 4.2.1 Predicting Velocities from Redshift Positions

Earlier in this section, we assumed access to the real-space positions, representing an idealized case. In practice, however, galaxy positions are measured in so-called *redshift* space, where the observed galaxy distances are shifted by an amount that depends on their peculiar velocities along the observer's line of sight. To bridge the gap between simulation data and real observations, we introduce a variant of our velocity prediction task that directly takes inputs as *redshift positions*. For simplicity, we

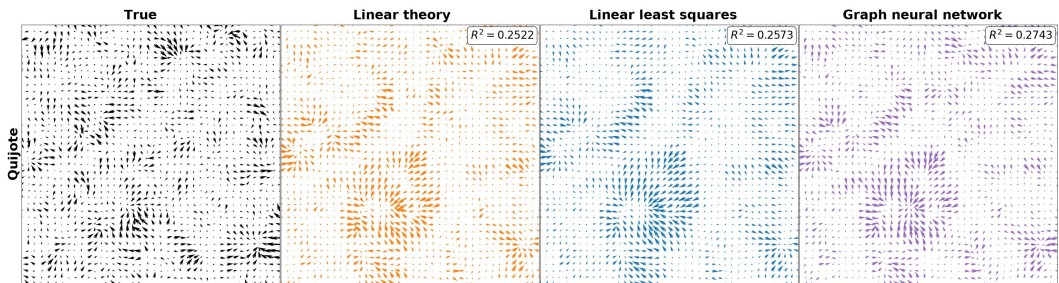

Figure 2: True vs predicted velocity fields for `Quijote`, projected and interpolated onto a 2D grid. Each arrow indicates the direction and magnitude of the field.

Table 3: Point Cloud Velocity Prediction. Time unit: 1-GPU time (or CPU for linear theory). Numbers marked with a $^*$ denote that additional "cosmological oracle" information was used.

| $\mathbf{R^2}\uparrow$ | Quijote | | | CAMELS-SAM | | | CAMELS | | |
|---|---|---|---|---|---|---|---|---|---|
| | **v** | Params. | Time | **v** | Params. | Time | **v** | Params. | Time |
| Linear theory oracle | 0.3769* | 1 | 12 sec | 0.2372* | 1 | 11 sec | 0.2970* | 1 | 6 sec |
| LLS | 0.4347 | 60 | 5 min | 0.2107 | 75 | 2 min | 0.2494 | 30 | 2 min |
| GNN | 0.4100 | 126k | 15 hr | 0.2865 | 126k | 1-3 hr | 0.2527 | 126k | 1-2 hr |

align the line of sight with the $z$-axis and replace the real-space positions $\{(x_i, y_i, z_i)\}_{i=1}^n$ with their redshift counterparts $\{(x_i, y_i, \tilde{z}_i)\}_{i=1}^n$, where $\tilde{z}_i = z_i \cdot (1 + \frac{v_{i,z}}{H_0})$, $v_{i,z}$ denotes the $z$-component of the velocity and $H_0 = 100\,h\,\mathrm{km\,s^{-1}\,Mpc^{-1}}$ denotes the "standardized" Hubble constant (since the galaxy positions are expressed in $h^{-1}\,\mathrm{Mpc}$). Note that the redshift effect is more notable at smaller scales. See more details and visualizations in App. B.3.

As shown in Tab. 4, replacing real-space positions with redshift-space positions leads to moderately lower performance for the original linear theory and LLS in `Quijote`, and substantial degradation at smaller scales. To mitigate such degradation, we propose simple modifications of these baselines (denoted as "Modified"), essentially shrinking the line-of-sight distance to counteract the redshift effect, which shows notable improvement in `CAMELS-SAM` and `CAMELS`; see details in App. B.3. In contrast, while the GNN exhibits similar performance drop to other baselines in `Quijote`, it performs better in `CAMELS-SAM` and `CAMELS` using redshift position inputs. This arises from GNN's ability to leverage the line-of-sight velocity information encoded in the redshift positions, effectively turning the original regression problem into a simpler denoising task, as evidenced by the per-axis performance reported in Tab. 11.

Table 4: Velocity Prediction from Redshift Positions. Numbers marked with a $^*$ denote that additional "cosmological oracle" information was used. For linear theory and LLS, each dataset column is split into results from the original baseline (see Sec. 4.2) and modified variants (see App. B.3).

| $\mathbf{R^2}\uparrow$ | Quijote | | CAMELS-SAM | | CAMELS | |
|---|---|---|---|---|---|---|
| | Original | Modified | Original | Modified | Original | Modified |
| Linear theory oracle | 0.3269* | 0.3269* | 0.1177* | 0.1621* | 0.0615* | 0.0732* |
| LLS | 0.3367 | 0.3362 | 0.1172 | 0.2116 | 0.0801 | 0.2185 |
| GNN | 0.3500 | | 0.3177 | | 0.3197 | |

## 5 Merger Tree Baselines and Results

### 5.1 Predicting Cosmological Parameters from Merger Trees

**1-Nearest Neighbor Predictor** Our first baseline is a simple nearest-neighbor model. This model considers, for each tree, the empirical distribution of individual features across all nodes and then evaluates a "distance" between trees by computing the Kolmogorov-Smirnov (KS) statistic [69] between empirical distributions. The cosmological parameters $(\Omega_{\mathrm{m}}, \sigma_8)$ of each test tree is predicted

by the parameters of the closest tree in the training set. More details are given in App. C.1. This model does not require any training, however each prediction for a test tree is expensive, and so we restrict the size of the training set to 2,000 trees and the size of the test set to 1,500 trees.

**DeepSets** To understand the role of the tree topology, we can simplify the merger tree as a *set* of node features (i.e., dropping all the directed edges). We then predict cosmological parameters via DeepSets [70] — a neural network for set inputs, defined as

$$
\text{DeepSet}(\{\mathbf{x}_i\}_{i=1}^n) = \rho\left(\sum_{i=1}^n \phi(\mathbf{x}_i)\right),
\tag{6}
$$

where $\rho, \phi$ are 2-layer MLPs with compatible dimensions. We use embedding size $d = 16$ and train DeepSets with Adam optimizer (batch size 128, maximum epochs 300).

**Graph Neural Networks** To fully exploit the tree topology, we use MPNNs [68] to process the directed tree, where directed edges flow from the ancestor halo nodes (i.e., progenitors) to the descendant nodes. The $l^{\text{th}}$ layer node embedding at node $i$ is computed as

$$
\mathbf{h}_i^{(l)} = \rho\left(\sum_{j \in \mathcal{N}(i)} \mathbf{h}_j^{(l)}\right), \quad \mathbf{h}_i^{(0)} = \mathbf{x}_i,
\tag{7}
$$

where $\rho$ is a 2-layer MLP, and the node neighbors $\mathcal{N}(i)$ consists of the ancestors of node $i$ and itself (i.e. self-loops are added). We train 4-layer MPNNs with the Adam optimizer same as DeepSets.

**Results and Analysis** We report the results on predicting cosmological parameters from a merger tree in the left panel of Tab. 5. We ablate the importance of node features, from individual features including mass $M$, concentration $c$, halo maximum circular velocity $v_{\text{max}}$, and scale factor $a$ (a measure of time), to all of them combined. Notably, the concentration is highly predictive of $\Omega_{\text{m}}$, whereas the scale factor $a$ is reasonably predictive of $\sigma_8$. Villaescusa-Navarro et al. [12] demonstrated that the internal properties of a single galaxy (mainly stellar mass, stellar metallicity, and maximum circular velocity) are sufficient to predict $\Omega_{\text{m}}$ with $\sim 10\%$ precision. Given that present-day galaxy properties encode their formation history, it is reassuring that a single merger tree provides even higher predictive power for $\Omega_{\text{m}}$. We find that GNNs offer an edge over DeepSets, showing the advantage of exploiting the tree topology. Smaller models (i.e. hidden size $d = 16$) slightly outperform larger on the validation set, likely due to the sparse nature of tree graphs where larger models can overfit.

## 5.2 Reconstructing Fine-Scale Merger Trees

**Extended Press-Schechter** Merger trees can be generated without a cosmological simulation via the Extended Press-Schechter (EPS) formalism [71, 72]. EPS provides a statistical framework for modeling the hierarchical assembly of halos by treating the growth of cosmic structures as a stochastic process, where the overdense regions (halos) effectively undergo a random walk. A merger is considered to occur when this random walk crosses a critical mass threshold, indicating the collapse of a region into a bound structure/halo. EPS starts with a halo at a given time, and probabilistically iterates back in time to construct the merger tree. However, EPS oversimplifies halo growth by assuming that all halos collapse ellipsoidally and that all time steps are uncorrelated. We use the EPS implementation by [73], where the EPS formalism is employed using Monte Carlo sampling, with corrections based on halo mass and shape. To obtain the EPS prediction for a given potentially unresolved merger node, we start the EPS algorithm with the features of the post-merger node (and cosmological parameters associated with the tree), and run the algorithm backwards in time. If the EPS algorithm predicts any split where both halos are above the minimum mass of the two pre-merger nodes, in the time set by the time-steps of the coarsened merger tree, we consider that a prediction of a merger. If not, we take that as the EPS prediction being no merger. Due to the stochastic nature of EPS, we average the predictions over 5 generated trees. We only test it on 20 test trees due to its slow sequential implementation; See App. C.2 for more discussion.

$k$**-Nearest Neighbor Classifier** We consider a simple node classifier via $k$-Nearest Neighbors (kNN), which only utilizes the 1-hop neighborhood information in the merger tree. Specifically, for each unresolved merger node $i$, we concatenate features from its post-merger node $\mathbf{x}_{\text{post}(i)}$ and its two pre-merger nodes $\mathbf{x}_{\text{pre1}(i)}, \mathbf{x}_{\text{pre2}(i)}$, and arrive at a feature vector $\tilde{\mathbf{x}}_i = [\mathbf{x}_{\text{post}(i)} \mid \mathbf{x}_{\text{pre1}(i)} \mid \mathbf{x}_{\text{pre2}(i)}]$.

For each tree, we perform hyper-parameter search over the number of neighbors $k \in [5, 25]$ on the validation set, and use the chosen $k$ to obtain the test set classification accuracy.

**Graph Neural Networks** We use the same architecture and training set-up as Sec. 5.1, except for changing the batch size as 1, i.e. taking a gradient step per tree; See App. C.2 for more details.

**Results and Analysis** As shown in Tab. 5 (right), all baselines perform better than random but have vast potential for improvements. We find that the scale factor $a$ is the more discriminative feature for merger node classification, based on which simple $k$-NN based on 1-hop neighbor information performs competitively to GNN with 4-hop message-passing. We also note that, unlike EPS, ML methods do not require knowledge of the cosmological parameters, yet still matching or surpassing it.

Table 5: Predicting cosmological parameters from merger trees on 24,966 `CS-Trees` (left) and classifying unresolved merger nodes on the largest 200 `CS-Trees` (right). Time unit: 1 GPU time (or CPU for EPS/1NN). Numbers with $*$ denote that additional cosmological information was used.

| Node Feat. | Baselines | CS-Trees (**R2**↑) | | | | | CS-Trees-200 (**Accuracy**↑) | | |
|---|---|---|---|---|---|---|---|---|---|
| | | $\Omega_{\mathrm{m}}$ | $\sigma_8$ | Params. | Time | | Merger Node Label | Params. | Time |
| $(M, a)$ | | | | | | EPS | 0.53 ±0.073* | – | 9 hr |
| $(c, v_{\mathrm{max}}, a)$ | 1NN | 0.64 ±0.063 | 0.31 ±0.112 | – | 4h49min | | | | |
| $M$ | | 0.12 ±0.009 | -0.03 ±0.009 | 0.61k | 10 min | | 0.61 ±0.005 | – | 12 sec |
| $c$ | | 0.68 ±0.008 | 0.21 ±0.010 | 0.61k | 10 min | | 0.53 ±0.005 | – | 12 sec |
| $v_{\mathrm{max}}$ | DeepSet | 0.57 ±0.010 | 0.14 ±0.012 | 0.61k | 10 min | $k$-NN | 0.59 ±0.005 | – | 12 sec |
| $a$ | | 0.23 ±0.012 | 0.48 ±0.010 | 0.61k | 10 min | | 0.72 ±0.005 | – | 12 sec |
| $(M, c, v_{\mathrm{max}}, a)$ | | 0.993 ±0.001 | 0.80 ±0.005 | 0.65k | 10 min | | 0.62 ±0.006 | – | 13 sec |
| $M$ | | 0.16 ±0.010 | -0.10 ±0.015 | 2.7k | 13 min | | 0.63 ±0.004 | 2.2k | 4 min |
| $c$ | | 0.84 ±0.004 | 0.35 ±0.010 | 2.7k | 13 min | | 0.61 ±0.005 | 2.2k | 4 min |
| $v_{\mathrm{max}}$ | GNN | 0.69 ±0.008 | 0.19 ±0.009 | 2.7k | 13 min | GNN | 0.61 ±0.005 | 2.2k | 4 min |
| $a$ | | 0.33 ±0.011 | 0.53 ±0.010 | 2.7k | 13 min | | 0.70 ±0.005 | 2.2k | 4 min |
| $(M, c, v_{\mathrm{max}}, a)$ | | 0.996 ±0.001 | 0.82 ±0.004 | 2.8k | 13 min | | 0.69 ±0.005 | 2.3k | 4 min |

# 6 Conclusion

We present COSMOBENCH, a collection of datasets from problems in cosmology designed to benchmark geometric deep learning. These datasets are curated from over 41 million core-hours of simulations and two petabytes of data, and they are currently the largest of their kind. Yet not all of these datasets are large by the standards of ML; many contain only hundreds or thousands of examples and nodes, particularly in `CAMELS`. Consequently, there are limits to purely data-driven or "brute-force" approaches, and there are also risks to overfitting with extremely large models. COSMOBENCH is an invitation to all researchers in ML, working on all types of models, to engage with this data and leverage the geometrical ideas and physical insights to produce enduring solutions in this space.

Looking ahead, we plan to expand COSMOBENCH with more data—at smaller scales and larger point clouds—to support development of deep architectures and foundation models in more data-rich regimes. Another promising direction is to utilize ML emulators to produce simulation data more efficiently, complementing traditional computationally-intensive simulation pipelines. To bridge the gap between simulations and observational data, we have taken the first step by including mock observations in redshift space for the velocity prediction task, and aim to further develop our benchmark with more survey realism and observational challenges.

**Acknowledgments and Disclosure of Funding**

The authors thank David Hogg (Flatiron), Soledad Villar (JHU), Jahmour Givansand (Flatiron), Natalí de Santi (UCB), Bruno Régaldo-Saint Blancard (Flatiron), Ruben Ohana (NVIDIA), Chirag Modi (Flatiron), and David Spergel (Flatiron) for valuable discussions, as well as Maya Bechler-Speicher (Meta), Michael Galkin (Google), and Christopher Morris (RWTH Aachen) for motivating this work. The authors also thank the anonymous NeurIPS reviewers and area chair for their constructive feedback. RS acknowledges financial support from STFC Grant No. ST/X508664/1, the Snell Exhibition of Balliol College, Oxford, and the CCA Pre-doctoral Program.

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

# A  Additional Dataset Details

## A.1  Cosmology Glossary

▷ **Cosmology**. The study of the Universe as a whole — its origin, structure, evolution, and fate — using physics, astronomy, and mathematics. In data-driven cosmology, simulations and observations are used to model the large-scale structure of the Universe.

▷ **Cosmological Simulations**. Cosmological simulations are large-scale computational models that simulate the evolution of the universe over billions of years. They use initial conditions based on physical laws and cosmological parameters to predict the formation of structures like galaxies, dark matter halos, and filaments.

▷ $N$**-body Simulations**. A type of cosmological simulation that models the gravitational interaction of a large number of particles (typically representing dark matter). These simulations are computationally intensive and key to studying the growth of structure over time.

▷ **Hydrodynamical Simulations**. Simulations that include both gravity and baryonic physics (e.g., gas dynamics, star formation, feedback). They model where galaxies form, how they evolve, and require solving coupled differential equations for both dark matter and fluid dynamics. Compared to the $N$-body simulations which only account for the force of gravity, the hydrodynamic simulations also solve the magneto-hydrodynamic equations and model astrophysical processes such as supernova and active galactic nuclei (AGN) feedback.

▷ **Dark Matter**. Dark matter is a form of matter that does not emit or absorb light, making it invisible. However, it makes up most of the universe's mass and is detectable only via its gravitational influence on galaxies and cosmic structure. It is a central component in cosmological models and simulations.

▷ **Dark Matter Halo**. A dark matter halo is a massive, invisible structure made mostly of dark matter that surrounds galaxies and galaxy clusters. Although it cannot be directly observed, its presence is inferred from its gravitational effects. Halos are the basic building blocks of cosmic structure. Galaxies form and nearly always live within dark matter halos.

▷ **Merger Tree**. A data structure that represents the hierarchical growth of a dark matter halo (and associated galaxies) over time. Each node is a halo, and branches show how smaller halos merged to form larger ones — analogous to a version-control tree of cosmic structure.

▷ **Semi-Analytical Models (SAMs)**. A computationally efficient approach that models galaxy evolution (processes like star formation, chemical enrichment, and black hole growth) as a set of coupled differential equations on top of the existing halo merger trees from $N$-body simulations. SAMs allow fast exploration of parameter space and are often used in combination with machine learning.

▷ **Large-Scale Structure**. The large-scale structure of the universe refers to the distribution of matter on scales of millions of light-years. It includes filaments, walls, voids, and galaxy clusters, forming a cosmic web. These patterns emerge naturally in simulations governed by gravity and initial density fluctuations.

▷ **Cosmological parameters**. A set of key numerical values that define the properties of the universe in a cosmological model. They include quantities like the matter density, expansion rate, amplitude of matter fluctuations.

▷ **Astrophysical parameters**. A set of numerical values that characterize the physical processes governing astrophysical objects, such as galaxies. These include parameters related to supernova feedback, active galactic nuclei, gas cooling, star formation, and other processes occurring within galaxies.

▷ $\mathbf{\Omega_m}$. The fraction of the total energy density of the universe made up of matter (including both dark matter and normal matter). If $\Omega_m$ is 1, the universe is matter-dominated and flat.

▷ $\mathbf{\sigma_8}$. A measure of how much matter has clumped together at a specific scale ($8~\mathrm{Mpc}/h$). It quantifies the amplitude of matter fluctuations, and is critical for modeling the growth of structure (like galaxies and clusters).

▷ $\mathbf{\Omega_b}$. The fraction of the universe's energy density made up of baryons, i.e., normal (non-dark) matter like protons and neutrons. It is a subset of $\Omega_m$.

- ▷ $n_s$. The spectral index of the primordial power spectrum. It describes how the initial density fluctuations vary with scale — whether small or large fluctuations were more prominent in the early universe.

- ▷ $h$. The dimensionless Hubble parameter defined as $h \equiv H_0/(100 \text{ km/s/Mpc})$ where $H_0$ is the current expansion rate of the universe. Units of length are often given normalized units of $\text{Mpc}/h$.

- ▷ **Mpc**. A megaparsec (Mpc) is a unit of distance used in astronomy equal to one million parsecs, or approximately 3.26 million light-years, or approximately $10^{22}$ meters.

- ▷ **cMpc**. A comoving megaparsec (cMpc) is a unit of distance equal to one megaparsec, measured in comoving coordinates. Comoving coordinates account for the expansion of the Universe, meaning that the distance between objects remains fixed in time if they are moving with the Hubble flow.

- ▷ **Concentration $c$**. An astrophysical parameter that describes how centrally dense a dark matter halo is. It is typically defined as the ratio of the halo's virial radius to its scale radius in a Navarro–Frenk–White (NFW) profile. Higher concentration values indicate that more mass is concentrated toward the center of the halo.

- ▷ $v_{\mathbf{max}}$. The maximum circular velocity of a halo or galaxy. The circular velocity is defined as the velocity of a circular orbit around the center of the halo at radius $r$.

- ▷ **Redshift $z$**. A measure of how much the wavelength of light has been stretched by the Universe's expansion, defined as $z = (\lambda_{\text{obs}} - \lambda_{\text{rest}})/\lambda_{\text{rest}}$, where $\lambda_{\text{obs}}$ is the observed wavelength and $\lambda_{\text{rest}}$ is the emitted/absorbed wavelength. In cosmology, redshift also serves as a measure of time, with $z = 0$ representing the present and $z \to \infty$ corresponding to the Universe's beginning.

- ▷ **Scale factor $a$**: A dimensionless quantity that describes the relative size of the Universe at a given time, related to redshift by $a = 1/(1 + z)$.

## A.2 Point Cloud Data Details

In this section, we provide more details of the point cloud datasets in COSMOBENCH. First, we give an overview of the cosmological simulations that produce point clouds of halos or galaxies. Then we explain in detail the simulation protocols used in `Quijote` [6], CAMELS-SAM [7], and CAMELS [8].

An $N$-body cosmological simulation models the universe by numerically evolving a large number of dark matter particles. Each simulation starts from initial conditions based on a particular set of cosmological parameters; it then tracks how these particles interact and move over time as the universe expands. The simulation produces a sequence of snapshots containing particle data (e.g. positions, velocities, and IDs), from which the halos (or galaxies) are identified using halo finder algorithms (typically via clustering particles). Each halo (or galaxy) becomes a point in the point cloud, with features such as its position, mass, velocity, and, concentration.

For the simulation suites in this work, the particles are initialized using second-order Lagrangian Perturbation theory[3] (2LPT): particles are assigned on a regular cubic periodic grid, and then *randomly* displaced and assigned peculiar velocities based on the amplitude and shape of the initial power spectrum. This initialization is stochastic—representing the cosmological uncertainty of the initial conditions of the Universe—meaning different point clouds were generated using different initial particle positions and velocities. The values of the cosmological parameters (and the astrophysical ones in the case of CAMELS) are varied over a broad range. The parameter space is sampled using either a Latin-Hypercube or a Sobol Sequence. For `Quijote`, the parameters varied are $\Omega_{\text{m}}, \Omega_{\text{b}}, h, n_s, \sigma_8$. For CAMELS-SAM and CAMELS, the only cosmological parameters varied are $\Omega_{\text{m}}$ and $\sigma_8$.

The initial conditions are generated at $z = 127$ for `Quijote` and CAMELS and at $z = 99$ for CAMELS-SAM. In the hydrodynamic simulations used in CAMELS, the initial conditions for two different fluids are generated using adiabatic initial conditions: dark matter and gas. The simulations are run to the present time ($z = 0$) and snapshots are stored at multiple redshifts from $z = 15$ to $z = 0$. The `Quijote` simulations use the Gadget-III code [74], while CAMELS-SAM and CAMELS use AREPO [75]. CAMELS employs the same subgrid physics model as the IllustrisTNG simulations.

---

[3] https://cosmo.nyu.edu/roman/2LPT/

From the simulation snapshots, the (halo or galaxy) point clouds are extracted by using the halo and subhalo finders (i.e., `Rockstar` [76] and `Subfind` [77]), which identify gravitationally bound structures based on the spatial distribution or the phase-space of the particles. The resulting point clouds respect periodic boundary conditions, inherited from the underlying $N$-body simulation. When a particle exits one side of the box, it re-enters from the opposite side, maintaining continuity across boundaries.

In what follows, we show sample point clouds obtained from (1) `Quijote` (top), (2) `CAMELS-SAM` (middle), and (3) `CAMELS` (bottom). We observe that as the box size decreases, the point distribution becomes increasingly sparse and irregular. The difference in point distribution arises from the cosmological parameters, the initial conditions of the simulation, and the tracer number density.

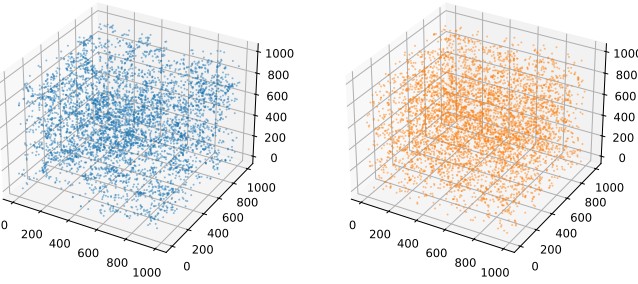

(1) `Quijote` clouds: (left) $\Omega_{\mathrm{m}} = 0.1003, \sigma_8 = 0.9504$, (right) $\Omega_{\mathrm{m}} = 0.4977, \sigma_8 = 0.9504$

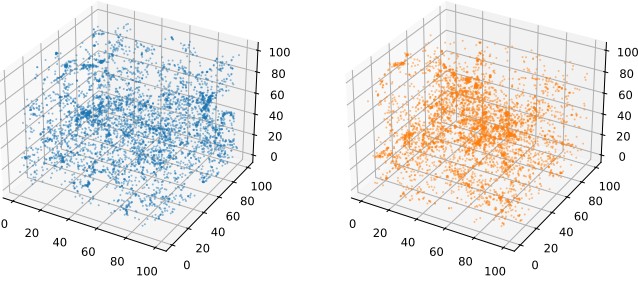

(2) `CAMELS-SAM` clouds: (left) $\Omega_{\mathrm{m}} = 0.3234, \sigma_8 = 0.6202$, (right) $\Omega_{\mathrm{m}} = 0.3238, \sigma_8 = 0.9982$

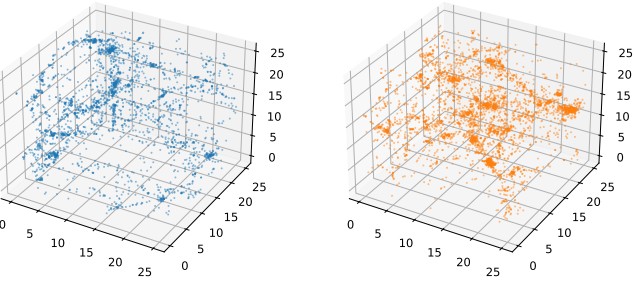

(3) `CAMELS` clouds: (left) $\Omega_{\mathrm{m}} = 0.4986, \sigma_8 = 0.6302$, (right) $\Omega_{\mathrm{m}} = 0.4994, \sigma_8 = 0.9966$

### A.3 Merger Tree Data Details

Merger trees for the `CAMELS-SAM` $N$-body simulations are generated in two steps: first, by identifying dark matter halos and subhalos from the particle data using `Rockstar`; and second by running these halo catalogs through `ConsistentTrees` [78] to construct the merger trees. The `Rockstar` algorithm uses an adaptive hierarchical refinement of *friends-of-friends* groups in seven dimensions (one in time; six in phase-space, positions and velocities). Briefly, friends-of-friends algorithms group two particles if they are within a given distance that scales by the mean particle separation in the initial conditions. The `ConsistentTrees` algorithm, especially created to pair with `Rockstar`, was built to explicitly ensure the consistency of halo properties like mass, position, and velocity, across time steps with a novel gravitational method. `ConsistentTrees` assumes that every halo has at least one progenitor in an earlier time step (logical if assuming large halos build by accretion and merging with other halos), and traces halos *backwards* in time to check for progenitors in previous timestep catalogs. See Figure 1 of Behroozi et al. [78] for a brief explanation of the algorithm.

This merger tree generation process can be done for any set of particle snapshots, but high temporal resolution leads to more physically realistic and useful trees. For example, the `ConsistentTrees` will give incoherent results if the backwards time steps are too large. Given that `Quijote` stored just 5 particle snapshots and the bulk of `CAMELS TNG` suite stored 35 to conserve storage space, merger trees generated from them do not have the sufficient resolution for cosmological applications. We thus only make use of the merger trees from `CAMELS-SAM` generated with 100 snapshots. This also motivates our task of reconstructing finer-scales merger trees from coarsely sampled ones.

Before formally describing merger trees, we recall some standard terminologies in graph theory. In a (directed) rooted tree, the *parent* of a node $n$ is the node connected to $n$ on the (directed) path to the root node; every node has a unique parent, except the root (which has no parent). A *child* node of a node $n$ is a node whose parent is $n$; each node may have one or more child nodes, except the leaf nodes (which have no children). For a binary tree, each node except the leaf nodes has exactly two child nodes.

Merger trees are directed trees, where the root node represents the halo at present time, and the directed edges describe how the progenitor (ancestor) halo evolves to the more recent halo. We call a node a *merger node* if it has more than one child nodes.

**Tree Trimming Procedure** To reduce storage requirement, condense information, and prevent information leakage, we trim the trees from `CAMELS-SAM` by first removing its subtrees with all nodes having mass less than $10^{10} M_\odot/h$ (i.e. those with fewer than 100 dark matter particles at all times, a rough guideline for a numerically well-resolved halo). We then prune each path in the resulting tree, where nodes with mass less than $3 \times 10^{10} M_\odot/h$ are removed, as illustrated in Fig. A.1. This trimming procedure also avoids potential information leakage, as the mass of the smallest halo (which always contains 20 particles) is proportional to $\Omega_m$.

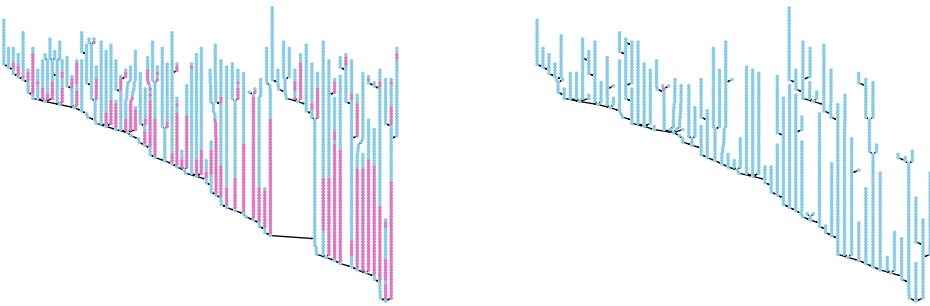

Figure A.1: Pruning the merger trees: given a tree (left), prune its paths by removing all nodes with mass less than $3 \times 10^{10} M_\odot/h$ (highlighted in red) and connecting the kept nodes within the same original path. This yields the pruned tree (right) used in `CS-Trees`.

**Tree Coarsening Procedure**  In Algorithm 1, we provide the full procedure of constructing the (binary) coarse-grained trees augmented with unresolved merger nodes for classification.

---

**Algorithm 1** Merger Tree Coarsening

---

**Require:** Tree $\mathcal{T}_0 = ([n_0], E_0, X_0)$ defined over full time steps $\{1, 2, \ldots, T\}$.
 1: **Pre-Binarization:** For each node with more than two pre-merger nodes in $\mathcal{T}_0$, retain only the two most massive nodes to obtain a binary tree $\mathcal{T}_{\text{bin}}$. Let $M_0$ be the set of merger nodes in $T_{\text{bin}}$.
 2: **Coarsening:**
 3:    Remove all nodes from even time steps $\{2, 4, \ldots, T-1\}$ in $\mathcal{T}_{\text{bin}}$ (retrain the root at time $T$).
 4:    Denote the set of kept nodes $[n']$ and features as $X'$.
 5:    Create a dictionary $D \leftarrow \{\text{parent}(m) : m \mid m \text{ is a removed node} \wedge m \in M_0\}$
 6:    **For each** $c \in [n']$, walk up to first kept parent node $p$ and add an edge $(c \rightarrow p)$ if not already visited.
 7:    Let the resulting tree be $\mathcal{T}_{\text{coarse}} = ([n'], E', X')$
 8: **Post-Binarization:** Apply binarization (see line 1) to $\mathcal{T}_{\text{coarse}}$ to obtain binary-coarsened tree $\mathcal{T} = (V, E, X)$
 9: **Virtual Node Augmentation:** Maintain the virtual node set $\tilde{V}$, virtual edges $\tilde{E}$, and labels $\mathbf{y}$.
10: **for** each $n$ in the set of merger nodes in $\mathcal{T}$ **do**
11:    Add virtual node $v(n)$ to $\tilde{V}$ (i.e., the unresolved merger nodes for classification)
12:    Add three virtual edges to $\tilde{E}$: $(n \rightarrow v(n)), (\text{child1}(n) \rightarrow v(n)), (\text{child2}(n) \rightarrow v(n))$.
13:    **if** $n \in \text{keys}(D)$ **then**
14:        Label $\mathbf{y}_{v(n)} \leftarrow 1$
15:    **else**
16:        Label $\mathbf{y}_{v(n)} \leftarrow 0$
17: **Return:** Binarized, coarsened tree $\mathcal{T}_c = (V + \tilde{V}, E + \tilde{E}, X \cup \mathbf{0}, \mathbf{0} \cup \mathbf{y})$ with virtual nodes and labels.

---

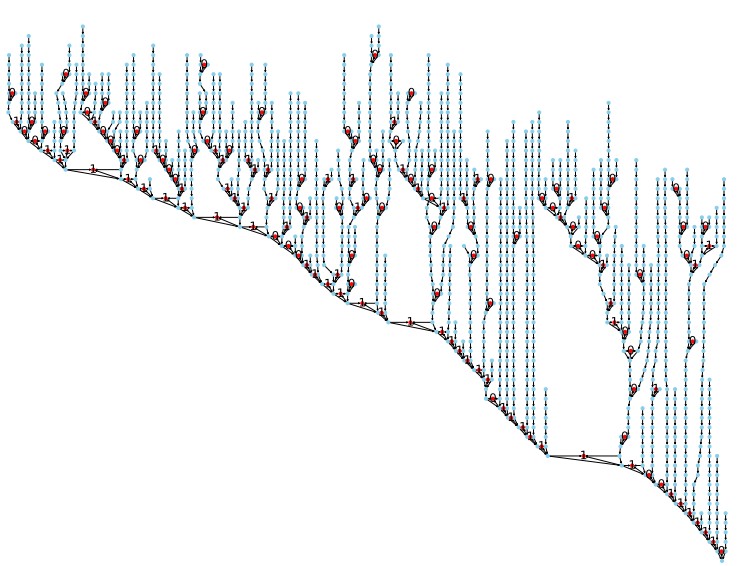

Figure A.2: An example coarsened merger tree. The coarsened tree has 1,362 nodes (blue dots) and 144 unresolved merger nodes (i.e., virtual nodes, red dots) with node labels attached (1 if there is a merger in the fine-grained tree, and 0 otherwise).

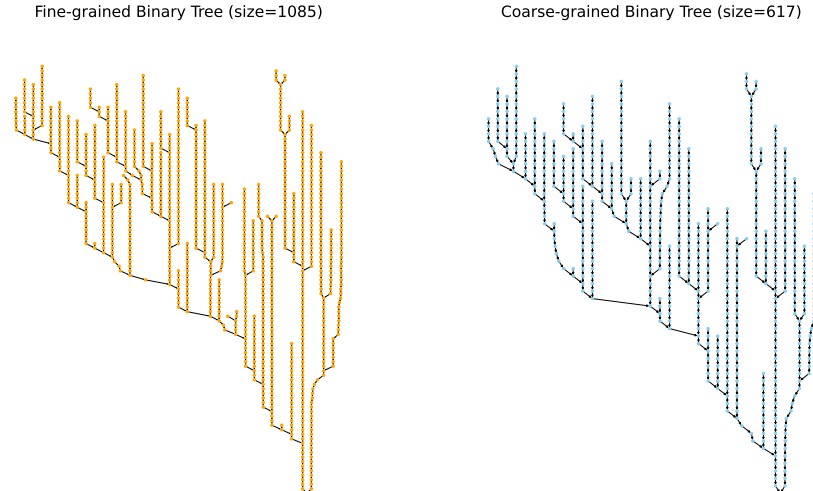

Figure A.3: Visual comparison of the fine-grained binary tree produced after Pre-Binarization in Algorithm 1 (left), and its corresponding coarse-grained binary tree produced after Post-Binarization in Algorithm 1 (right).

# B  Additional Point Cloud Experimental Results

## B.1  Predicting Cosmological Parameters from Point Clouds

**2PCF Definition**  The two-point correlation function quantifies the excess probability $\mathrm{d}P$ of finding a pair of points separated by a 3D distance $r$, relative to a uniform random distribution,

$$\mathrm{d}P = n\left[1 + \xi(r)\right]\mathrm{d}V, \tag{8}$$

where $n$ is the number density of points and $\mathrm{d}V$ is the infinitesimal volume element. We use `Corrfunc` [53] to compute the two-point correlation function, using the Landy & Szalay estimator [52] defined as

$$\xi_i = \frac{1}{\mathrm{RR}_i}\left[\mathrm{DD}_i\left(\frac{n_R}{n_D}\right)^2 - 2\mathrm{DR}_i\left(\frac{n_R}{n_D}\right) + \mathrm{RR}_i\right], \tag{9}$$

where $\mathrm{DD}_i$, $\mathrm{DR}_i$, and $\mathrm{RR}_i$ are the counts of data–data (halo or galaxy), data–random, and random–random pairs, respectively, in the $i^{\mathrm{th}}$ bin of radial separation. Here, $n_{\mathrm{D}}$ and $n_{\mathrm{R}}$ denote the number density of data and random points used.

For all datasets, we adopt logarithmic radial binning. For the `Quijote` halos, simulated in a 1000 Mpc/$h$ box, we use 25 bins spaced logarithmically between 0.5 and 120 Mpc/$h$. For the `CAMELS-SAM` galaxy catalog, extracted from a 100 Mpc/$h$ box, we use 25 bins spanning 0.0125 to 12 Mpc/$h$. For the higher-resolution `CAMELS` sample, which resides in a 25 Mpc/$h$ box, we use 25 bins between 0.0125 and 3 Mpc/$h$. To compute $\mathrm{DR}, \mathrm{RR}$ in Equation (9), we use a random point cloud with 100 times more points than the data. The $\{\xi_i\}$ then become the input features for the MLP, as we discuss next.

**MLP on 2PCF Training Details**  For the two-point correlation function cosmology predictions, we construct a simple 4-layer MLP, with tunable hyperparameters. When we pass our two-point correlation function to the MLP, we handle the occasional unphysical (small) negative values at large radii due to shot noise by taking their absolute values. We further scale the correlation features logarithmically to reduce their data range, especially as the values for `CAMELS-SAM` and `CAMELS` at lower bins are significantly high. The hidden dimensions of the first and second layers range $[64, 128]$, and the third layer from $[16, 64]$, with the final layer leading to the prediction of $[\Omega_{\mathrm{m}}, \sigma_8]$. We logarithmically tune the learning rate in the range $[10^{-5}, 10^{-2}]$, and apply dropout with a rate chosen from the interval $[0.0, 0.5]$. The batch sizes range from $\{4, 16, 64\}$, and the model is trained for a fixed number of 300 epochs. We report the test results from the model selected based on the best validation performance, tuned over 100 trials using the tree-structured Parzen Estimator sampler [79, 80].

**MLP on 2PCF Ablation**  We investigate the effect of different 2PCF binning choices: (i) the scale and number of bins for a fixed bin range; (ii) the bin range for fixed scale and number of bins. Ablation (i) is carried out for `Quijote` using the same bin range between 0.5 and 150 Mpc/$h$, with results shown in Tab. 6. We observe that logarithmic scale is better than linear scale, especially for predicting $\sigma_8$ with 25 bins, as it better resolves small distances. Fitting 2PCF-MLP on a logarithmic scale with 25 bins (first row in Tab. 6) produces a test Mean Squared Error (MSE) of $(2.10 \pm 0.04) \times 10^{-3}, (2.16 \pm 0.04) \times 10^{-3}$ for predicting $\Omega_{\mathrm{m}}, \sigma_8$, respectively. Balla et al. [43] reported a test MSE of $(2.03 \pm 0.02) \times 10^{-3}, (4.66 \pm 0.06) \times 10^{-3}$ using the same logrithmic 25 bins and a larger MLP model on a subset of `Quijote` point clouds (2048/512/512 training/validation/test split). In comparison, using our full `Quijote` set (19,651/6,551/6,550 training/validation/test split) produces similar performance on predicting $\Omega_{\mathrm{m}}$ while notably better performance on predicting $\sigma_8$. We skip log feature normalization for Ablation (i) because the `Quijote` correlation values are within a small range unlike `CAMELS-SAM` and `CAMELS`, and we observe no difference by applying log normalization. Ablation (ii) is conducted on all three datasets by comparing 25 logarithmic bins using $R_{\min} = \frac{B}{2000}, R_{\max} = \frac{B}{25/3}$ where $B$ is the box size (denoted as "Base"), with their counterparts that either reduce the minimal bin size four times (denoted as "Base $\times \frac{R_{\min}}{4}$"), or quadruple the maximum bin size (denoted as "Base $\times 4R_{\max}$"). We report the results in Tab. 7, which shows the effect of the bin range on the performance. We use the choice of "Base $\times 4R_{\max}$" for `Quijote` and `CAMELS`, and "Base $\times \frac{R_{\min}}{4}$" for `CAMELS-SAM` in the main text Tab. 2.

Table 6: 2PCF Binning Ablation for Point Cloud Cosmological Parameter Regression ($R^2 \pm$ 1std): (i) The Effect of Scales and Number of Bins

| $\mathbf{R^2} \uparrow$ | Bin Choice | | Quijote | |
|---|---|---|---|---|
| | scale | bins | $\Omega_m$ | $\sigma_8$ |
| 2PCF | log | 25 | 0.84 ±0.004 | 0.84 ±0.004 |
| | linear | 25 | 0.83 ±0.004 | 0.74 ±0.006 |
| | log | 250 | 0.83 ±0.004 | 0.84 ±0.004 |
| | linear | 250 | 0.83 ±0.004 | 0.83 ±0.004 |

Table 7: 2PCF Binning Ablation for Point Cloud Cosmological Parameter Regression ($R^2 \pm$ 1std): (ii) The Effect of Bin Range

| $\mathbf{R^2} \uparrow$ | Bin Choice | Quijote | | CAMELS-SAM | | CAMELS | |
|---|---|---|---|---|---|---|---|
| | | $\Omega_m$ | $\sigma_8$ | $\Omega_m$ | $\sigma_8$ | $\Omega_m$ | $\sigma_8$ |
| 2PCF | Base $\times \frac{R_{min}}{4}$ | – | – | 0.73 ±0.03 | 0.82 ±0.02 | – | – |
| | Base | 0.84 ±0.004 | 0.83 ±0.004 | 0.64 ±0.03 | 0.74 ±0.03 | 0.85 ±0.02 | 0.30 ±0.06 |
| | Base $\times 4R_{max}$ | 0.85 ±0.004 | 0.84 ±0.004 | 0.65 ±0.04 | 0.78 ±0.03 | 0.84 ±0.02 | 0.30 ±0.06 |

**LLS Ablation**  We investigate the effect of the cutoff radius in LLS, which are selected using the validation set to greedily find the 12 best cutoff thresholds given a candidate threshold set, and chosen separately for each target parameter ($\Omega_m, \sigma_8$). We now provide another LLS baseline on a "naive" set of cutoff features, which are chosen from the candidate set with equi-spaced elements, and applied to both parameters. As shown in Table 8, this naive choice of cutoff features (first row) yields a poorer fit from the linear model than the optimized features used in the main paper (second row) and performs worse than GNNs (third row) in CAMELS-SAM for both parameters and CAMELS for $\Omega_m$.

Table 8: The Effect of LLS Features for Cosmological Parameter Regression

| $\mathbf{R^2} \uparrow$ | Quijote | | CAMELS-SAM | | CAMELS | |
|---|---|---|---|---|---|---|
| | $\Omega_m$ | $\sigma_8$ | $\Omega_m$ | $\sigma_8$ | $\Omega_m$ | $\sigma_8$ |
| LLS (naive) | 0.83 ±0.004 | 0.80 ±0.004 | 0.68 ±0.04 | 0.73 ±0.03 | 0.74 ±0.03 | 0.26 ±0.06 |
| LLS | 0.83 ±0.004 | 0.80 ±0.004 | 0.77 ±0.03 | 0.82 ±0.02 | 0.78 ±0.03 | 0.28 ±0.06 |
| GNN | 0.80 ±0.004 | 0.77 ±0.005 | 0.75 ±0.03 | 0.83 ±0.02 | 0.78 ±0.03 | 0.24 ±0.06 |

**GNN Message-Passing Details**  Our higher-order graphs are minimal examples of combinatorial complexes, which enables modeling of higher-order relations via the introduction of hierarchies between higher-order cells (subset of the point cloud with more than two nodes, or higher than binary relations), while retaining flexibility through allowing set-type relations [64]. By using cells including galaxies or halos at different distance scales, one is able to effectively abstract information at diverse scales. Higher-order message-passing thus facilitates modeling long-range information more efficiently, compared to standard message-passing (between nodes via edges).

The combinatorial complex is a generalization of graphs, defined as $(S, \chi, \text{rank})$, where $S$ is a set (points in the point cloud), $\chi$ is a skeleton (set of all cells), and rank$(\cdot)$ maps each cell into its rank in nonnegative integers ($k = 0 \ldots n$). In our experiments, we use tetrahedral cells constructed from a 3-dimensional Delaunay triangulation to identify edge adjacency structure. We also perform ablation study using standard graph with node adjacency (edges) only. See Fig. B.1 for an illustration.

From the perspective of combinatorial complex, we can generalize our GNN message-passing scheme in Equation 2 as follows:

$$\mathbf{m}_z^{(l)} = \sigma \left[ \bigotimes_{k=0}^{n} \bigoplus_{y \in \mathcal{N}_k(z)} \psi_{\mathcal{N}_k, \text{rank}(z)}(\mathbf{h}_y^{(l)}, \text{Inv}(\mathbf{x}_z, \mathbf{x}_y)) \right]. \tag{10}$$

Here, we employ multiple trainable non-linear functions $\psi$, for each type of neighborhood and rank of $z$. For example, $\psi_{ne}$ in Equation 2 indicates the function tied to the message-passing from nodes to edges $z \in E$. Thus, $\psi_{\mathcal{N}_k, \text{rank}(z)}$ is a generalized form applied to messages passed from cells with rank of k to cells with the same rank as $z$.

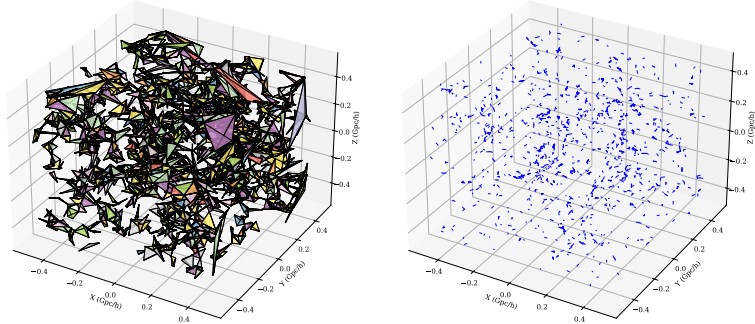

Figure B.1: Illustration of the graphs constructed from one example point cloud in `Quijote`: (left) the higher-order graph with tetrahedral cells to identify edge neighbors; (right) the standard graph.

Similar to GNNs, different types of trainable non-linear functions $\psi$, including attention-based algorithms, can be utilized. In this study, we modify convolutional push-forward and merge-node operations for combinatorial complexes introduced in [64, 81, 82]. Briefly, convolutional push-forward operations generate $l$-th layer messages $K_{i \to j}^{(l)} \in \mathbb{R}^{N_j \times d_{\text{in}}}$ from rank $i$-cells to $j$-cells, as $K_{i \to j}^{(l)} = G H_i^{(l)} W$, using a neighborhood matrix $G \in \mathbb{R}^{N_j \times N_i}$, feature matrix $H_i^{(l)} \in \mathbb{R}^{N_i \times d_{\text{in}}}$, and trainable matrix $W \in \mathbb{R}^{d_{\text{in}} \times d_{\text{out}}}$. For more flexibility, we generalize the convolution push-forward operation to mimic multiple aggregation schemes by relaxing the original formula into $K_{i \to j}^{(l)} = G * (H_i^{(l)} W)$, with $A * B = \bigoplus_l A_{kl} B_{lm}$, in Einstein summation convention. The operator $\bigoplus_l$ is a intra-neighborhood, permutation-invariant aggregation function as defined in Equations 2 and 10. Finally, the merge-node operations are simple aggregation of messages $K_{i \to j}^{(l)}$ across different neighborhoods, or in this case, $M_j^{(l)} = \bigotimes_{i=0}^n K_{i \to j}^{(l)}$. Again, $\bigotimes$ denotes the inter-neighborhood aggregation function.

**GNN Training Details**    For training the GNNs defined in (2), we perform hyper-parameter search across 100 configurations, including the cutoff radius $R_c \in [0.01, 0.015, 0.02]$ controlling the sparsity of the graph, the types of E(3)-invariant features (Euclidean, Hausdorff, None), the number of GNN layers $L \in \{1, \ldots, 6\}$, and the GNN hidden dimensions from $\{32, 64, 128, 256\}$. We explore different intra- and inter-aggregation functions (i.e., sum, max, min, and all three combined) and nonlinear activation functions (tanh or ReLU). For datasets `CAMELS-SAM` and `CAMELS`, we limit the batch size to smaller values ($\{1, 2, 4, 8\}$), while for the `Quijote` we include batch sizes up to 16. We also search over the learning rate $[10^{-5}, 10^{-3}]$, weight decay $[10^{-7}, 10^{-5}]$, neighborhood dropout probability $[0, 0.2]$, and the number of epochs for cosine annealing ($T_{\max} \in \{10, 100\}$). Each trial consists of 300 epochs, and we report our test results based on the best validation results. We conduct 100 trials for both `CAMELS-SAM` and `CAMELS`, whereas for `Quijote`, we carry out 25 trials due to computational limitations.

**GNN Ablation**    We perform an ablation study to see the impact of edge-edge message-passing defined on the higher-order graphs. We adopt the GNN (w/o edgeMP) model by disabling message-passing between edges defined on the parent tetrahedron (see Fig. B.1 as a comparison). We observe marginal differences in the performance metrics mostly within the 1 std, as shown in Tab. 2 and Fig. B.2. This may arise from multiple factors, including optimization challenges introduced by increased model complexity, or the possibility that the constructed higher-order graphs do not offer structural advantages over simpler ones. As a proof of concept, our higher-order graphs operate on tetrahedral cells identified from the Delaunay triangulation. We note that such tetrahedral cells are still locally confined, not modeling much larger scales compared to edges in standard graphs. Incorporating hyper-edges or clusters beyond edges or tetrahedral cells may further improve the performance on cosmological parameter prediction as shown recently in [63]. Our code base[4] allows exploration to even higher-orders involving a large number of galaxies and halos, at diverse scales.

---

[4] https://github.com/Byeol-Haneul/CosmoTopo/tree/benchmark

In future work, we aim to improve the construction of combinatorial complexes adapted to these cosmological point clouds for better performance.

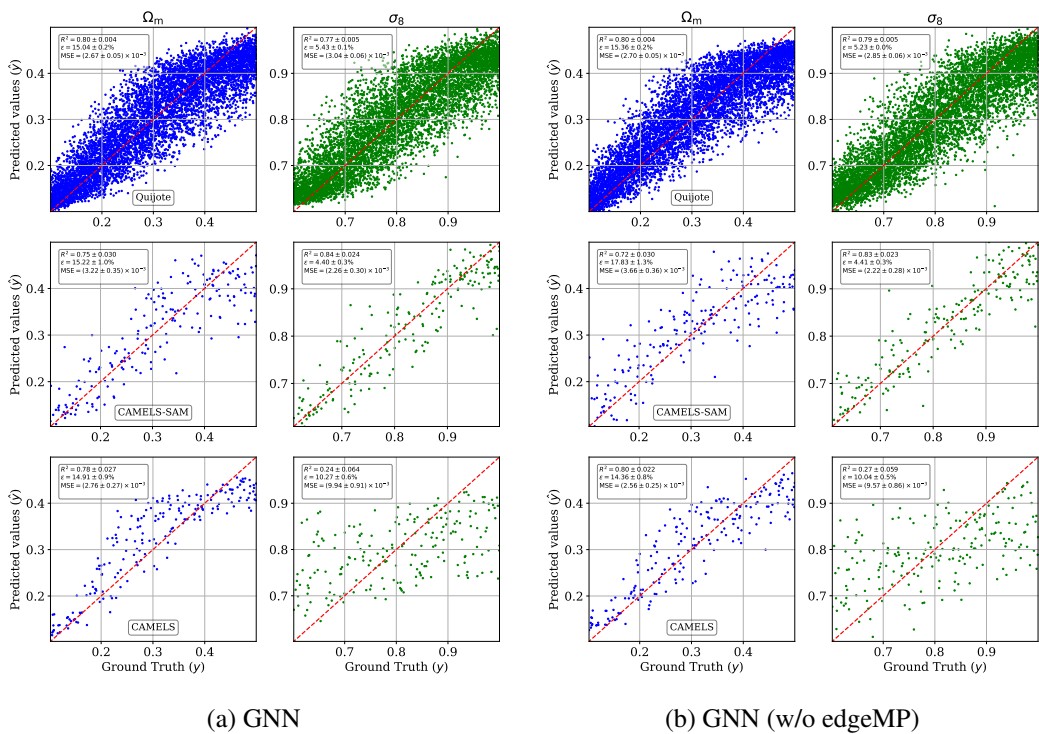

(a) GNN                                          (b) GNN (w/o edgeMP)

Figure B.2: Cosmological parameter predictions using (a) GNN on higher-order graphs (with both node and edge message-passing); (b) GNN (without edge message-passing), for test sets in `Quijote` (top), `CAMELS-SAM` (middle), and `CAMELS` (bottom), respectively.

**Discussion of Results**    Our results reported in Tab. 2 (with further details in Fig. B.2) show that ML baselines are comparable to 2PCF baseline for `Quijote` and `CAMELS`, while being strictly better for `CAMELS-SAM`. A promising next step for a more comprehensive evaluation is to test these baselines on observational data which include experimental noise.

**Predicting Additional Parameters.**    For our baseline tasks, we focus on the two cosmological parameters $\Omega_{\mathrm{m}}, \sigma_8$ due to their significance in astrophysics (they are among the least well-constrained cosmological parameters from the observational data), as well as the feasibility of predicting them from the positions of halos and galaxies (other parameters such as $\Omega_{\mathrm{b}}, n_s, h$ are more difficult to measure from positions alone and typically require information such as the cosmic microwave background or supernovae). Nonetheless, we are interested to see how well ML can predict additional parameters and bridge the gap to real-world inference problems. A shown in Tab. 9, these baselines perform significantly worse on predicting $\Omega_{\mathrm{b}}, n_s, h$ compared to $\Omega_{\mathrm{m}}, \sigma_8$, as expected.

Table 9: Cosmological Parameter Regression using `Quijote`: $\Omega_{\mathrm{b}}$, $n_s$, and $h$.

| $\mathbf{R}^2 \uparrow$ | $\Omega_{\mathrm{b}}$ | $n_s$ | $h$ |
|---|---|---|---|
| 2PCF | 0.29±0.009 | 0.23±0.009 | 0.21±0.008 |
| LLS | 0.19±0.008 | 0.21±0.009 | 0.20±0.008 |
| GNN | 0.07±0.006 | 0.08±0.008 | 0.14±0.009 |

## B.2 Predicting Velocities from Positions

**Linear Theory Computation**  Following the discussion Sec. 4.2, the linearized continuity equation implies that

$$\nabla \cdot \boldsymbol{v}(\boldsymbol{x}) = -aHf\,\delta(\boldsymbol{x}), \tag{11}$$

where $a$ is the scale factor, $H = \dot{a}/a$ is the Hubble parameter, $f$ is the linear growth rate (which is a function of the cosmological parameters). Furthermore, in linear theory, the overdensity of halos (or galaxies) $\delta_h$ is related to the overdensity of matter $\delta$ via linear rescaling by a constant $b$, such that $\delta_h = b\,\delta$. (11) can thus be written in Fourier space, yielding the velocity in Fourier space $\boldsymbol{v}(\boldsymbol{k})$, as

$$\boldsymbol{v}(\boldsymbol{k}) = -i\frac{aHf}{b}\,\frac{\boldsymbol{k}}{k^2}\,\delta_h(\boldsymbol{k}), \tag{12}$$

where $\boldsymbol{k}$ is the Fourier conjugate of the position $\boldsymbol{x}$, $k \equiv |\boldsymbol{k}|$ is the magnitude of $\boldsymbol{k}$, and $\delta_h(\boldsymbol{k})$ is the Fourier transform of the halo overdensity. Since the linear velocity field is irrotational (its curl is zero), its Fourier components align with $\boldsymbol{k}$. The quantity $aHf/b$ is the proportionality constant discussed in the main text. We refer the reader to [65] for further information on cosmological linear theory or to [14] where this is used to predict velocities from the observed distribution of galaxies.

While Equation (3) provides an integral solution in position space (i.e. $\boldsymbol{x}$ space), the Fourier exposition of Equation (12) provides a practical, commonly used, way to infer the velocity field from the density field. We first compute $\delta_h(\boldsymbol{x})$ by computing the halo density on a grid of size $N_g$ (effectively by counting the number of halos in each cell, and interpolating using a triangular-shaped cloud scheme). We then Fourier transform to obtain $\delta_h(\boldsymbol{k})$. The value of $\boldsymbol{k}$ on each Fourier grid point is given as $2\pi/L(i,j,k)$, where $i,j,k \in \{0,1,...N_g - 1\}$. We choose the value of $N_g$ that gives the best $R^2$ on the velocity prediction for the validation set, considering values between $N_g = 10$ and 90 in increments of 10. Up to the proportionality constant, we evaluate $\boldsymbol{v}(\boldsymbol{k})$ and then Fourier transform back to obtain $\boldsymbol{v}(\boldsymbol{x})$. All these computations were performed using the `Pylians` package[5].

The proportionality constant, $aHf/b$, depends on the cosmological parameters and varies from point cloud to point cloud. Thus, for our "linear theory oracle" result, we fit for this proportionality constant (as well as for $N_g$) on each point cloud while minimizing the $R^2$ on the $\boldsymbol{v}(\boldsymbol{x})$ prediction. This gives the upper limit of what can be achieved with linear theory and a perfect knowledge of the cosmological parameters (which could be achieved from a different cosmological dataset, such as the cosmic microwave background [65]), thus serving as an oracle to establish a baseline for the other predictions. On the one hand, the oracle knows extra information about the cosmological parameters, potentially boosting the velocity prediction accuracy, while on the other hand, it relies on linear theory which is known to break down on small scales and thus produce inaccurate results.

**LLS Ablation**  We find the results of LLS depends crucially on using the ensembles of features (inverse of powers of pairwise distances for all power less or equal to $P$). In Tab. 10 we run LLS with a single choice of power order $p$ at a time (where $K = 10$). We see that the performance of LLS on a single power order is not superior to GNN.

Table 10: The Effect of LLS Features for Velocity Prediction on `Quijote`

| Model | LLS (P=1) | LLS (P=2) | LLS (P=3) | LLS (P=4) | LLS ($1 \leq P \leq 4$) | GNN |
|---|---|---|---|---|---|---|
| $R^2 \uparrow$ | 0.408 | 0.404 | 0.385 | 0.000 | 0.435 | 0.410 |

**GNN Training Details**  We use hidden dimension size $d = 64$ and perform hyper-parameter search over the number of message-passing layers $L \in [4, 5, 6]$. All training is done using Adam optimizer with batch size 1 (gradient update per cloud), learning rate 0.001, and a maximum of 300 epochs. We report the test set performance on the best model selected based on the validation set performance.

**Discussion of Results and Additional Visualizations**  To visualize the predicted velocities of the point clouds, as in Fig. 2, we compute the velocities on a 3D grid, and then project along one dimension. We use the `yt` package[6]. Fig. 2 shows results for `Quijote`, with the true velocities in the

---

[5]https://pylians3.readthedocs.io/
[6]https://yt-project.org/

left panel, followed by the linear-theory oracle, LLS, and GNN from left to right. We observe that the true velocity field contains many complex flow paths, while the linear theory and LLS predictions tend to predict smoother flows, typically following the large-scale bulk motion. This is expected due to their linear nature. The GNN also produces smoother predictions than the truth, but it picks up on some of the more smaller-scale flows. Take Fig. 2 as an example: the GNN predicts a small local sink towards the lower middle of the panel, while the linear predictions have all flow lines going in the same direction towards a larger sink.

For further visualization, Fig. B.3 provides a similar plot to Fig. 2, but for two different test set point-cloud examples from each of the multiscale simulations (Quijote, CAMELS-SAM, and CAMELS). We additionally report the $R^2$ for the specific point cloud being plotted in the top right corner of each panel. The trends for the two Quijote examples (first and fourth rows) are similar to Fig. 2, although it is interesting to note that the fourth row is an example where LLS is better than the GNN. CAMELS-SAM and CAMELS produce much smoother true velocities in the image, as they are far smaller scale systems than Quijote, thus the chosen grid resolution traces the velocities more smoothly. For CAMELS-SAM, the two examples (second and fifth rows) are consistent with the average behavior reported in Tab. 3, with the GNN performing best, while for CAMELS the linear theory oracle performs the best. The most accurate performance of the linear theory oracle for CAMELS is a notable result, as linear theory is known to break down on the small scales probed by CAMELS, but nonetheless, this implies that a linear relation is sufficient to reasonably predict the velocity (more accurately so compared to the other methods currently in this benchmark) if one exactly knew the constant of proportionality, as our oracle does. A future entry to the benchmark could use linear, or even non-linear, theory without any oracle-like information by performing the non-trivial task of jointly inferring the density, cosmological parameters, and in turn the velocity via Bayesian hierarchical modeling (see e.g. [66, 67, 83, 84] for works in this direction).

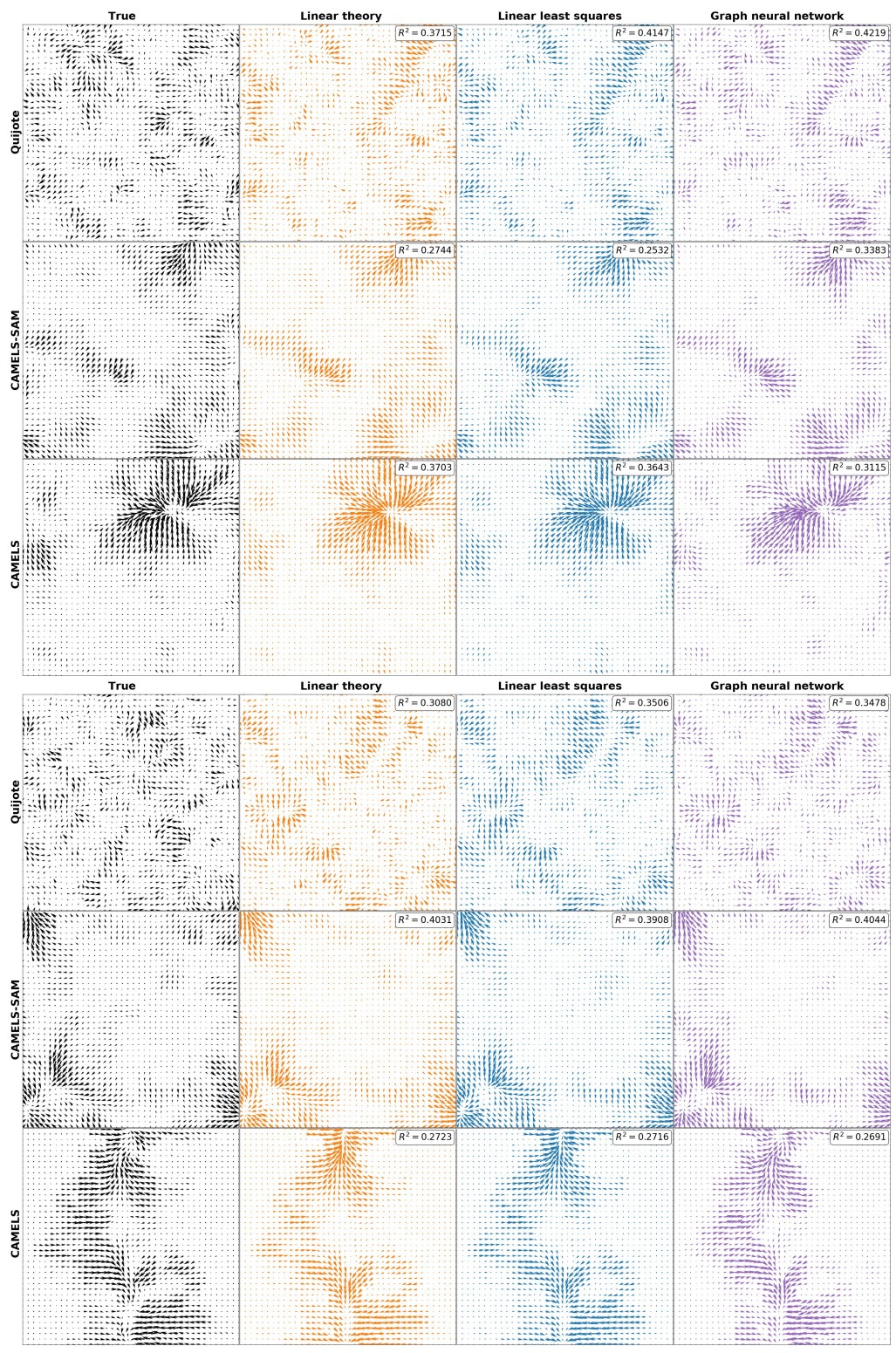

Figure B.3: True (left column) vs predicted (right three columns) velocity fields for two random example point clouds from each simulation suite (e.g., two examples from `Quijote` are shown in the top row and the fourth row). The velocities are projected and interpolated onto a 2D grid using the `yt` package. Each arrow indicates the direction and magnitude of the velocity at that position. The $R^2$ for each example is shown in the top right corner of each panel.

## B.3 Predicting Velocities from Redshift Positions

**Redshift Data Details**   In cosmology, the redshift position vector $s \in \mathbb{R}^3$ is obtained by displacing the real-space position $x \in \mathbb{R}^3$ along the line of sight proportional by the object's peculiar velocity,

$$s = x + \frac{v^\top \hat{n}}{aH}\hat{n}, \tag{13}$$

where $a \equiv a(t)$ is a dimensionless scalar describing the time $t$, $H$ is the Hubble parameter measuring the universe expansion rate, and $\hat{n}$ is the line-of-sight unit vector. For our simulation point clouds modelling the present-day universe, we have $a = 1$ and $H \equiv H_0 = 100\,h\ \mathrm{km\,s^{-1}\,Mpc^{-1}}$. For simplicity, we align the line of sight direction with the $z$-axis, namely $\hat{n} = [0, 0, 1]^\top$. Thus we arrive at the redshift position along the $z$-axis as $\tilde{z} = z \cdot (1 + \frac{v_z}{100})$, where the redshift positions along the $x$-axis and $y$-axis remain the same as the real-space positions. Fig. B.4 provides visualizations of the 2D real-space positions $(x, z)$ on the left panel and 2D redshift positions $(x, \tilde{z})$ on the right panel. We note that the redshift effect is more significant at smaller scales (e.g., CAMELS) than large scale (e.g., Quijote).

**Linear Theory Modification Details**   Recall in real space, linear theory relates the overdensity of halos (or galaxies) $\delta_h$ to the overdensity of matter $\delta$ linearly, $\delta_h = b\,\delta$. In redshift space this is modified by the Kaiser effect [85], such that $\delta_h = (b + f\mu^2)\delta$, where $f \approx \Omega_m^{0.55}$ and $\mu$ is the cosine of the angle to the axis of redshift distortion (here the $z$-axis). This modification essentially stretches the field to account for the redshift-space distortions. For our linear theory *oracle* we treat $b$ and $f$ as free parameters which are fit to the data.

**LLS Modification Details**   For the modified variant of LLS, we scale the $z$-axis distance by a shrinkage factor $\lambda$, where $\lambda$ is searched over $\{0.95, 0.99\}$ for Quijote, and $\{0.3, 0.4, 0.5, 0.6\}$ for CAMELS-SAM and CAMELS using the validation set. Note that $\lambda = 1.0$ recovers the original LLS baseline.

**Discussion of Results**   As shown in Tab. 11, all baselines (including their modifications, if applicable) perform similarly in Quijote, with notable worse performance on predicting $v_z$ affected by the redshift line of sight than predicting $v_x, v_y$. Applying modifications in linear theory and LLS boosts their performance in CAMELS-SAM and CAMELS [7]. Surprisingly, GNN performs better in CAMELS-SAM and CAMELS when replacing real-space positions with redshift positions, due to the improvement along the line-of-sight $z$-axis. This arises from a simplified learning objective. Recall the redshift position input is a "noisy" version of the output, given by $\tilde{z} = z \cdot (1 + \frac{v_z}{100})$. Therefore the GNN can learn to solve a simpler denoising task, rather than fitting an arbitrary map from positions (input) to velocities (output). This is supported by our ablation experiment, where we choose the line-of-sight direction as the $x$-axis and train the same GNN model from scratch. We find per-axis $R^2$ in CAMELS as $R^2(v_x) = 0.5011$, $R^2(v_y) = 0.2155$, $R^2(v_z) = 0.2215$, demonstrating the notable boost along line-of-sight $x$-axis compared to the other axes.

Table 11: Velocity Prediction from Redshift Positions. Column header "ALL" denotes overall $R^2$ across the three axes, whereas "$v_x$", "$v_y$" "$v_z$" denote per-axis $R^2$. Numbers marked with a * denote that additional "cosmological oracle" information was used.

| $R^2$ ↑ | Quijote | | | | CAMELS-SAM | | | | CAMELS | | | |
|---|---|---|---|---|---|---|---|---|---|---|---|---|
| | All | $v_x$ | $v_y$ | $v_z$ | All | $v_x$ | $v_y$ | $v_z$ | All | $v_x$ | $v_y$ | $v_z$ |
| Linear theory oracle | 0.3269* | 0.3629* | 0.3629* | 0.2527* | 0.1177* | 0.2401* | 0.2390* | -0.1268* | 0.0615* | 0.1653* | 0.1564* | -0.1467* |
| w/ modification | 0.3269* | 0.3628* | 0.3628* | 0.2527* | 0.1621* | 0.2902* | 0.2891* | -0.0938* | 0.0732* | 0.1959* | 0.1840* | -0.1686* |
| LLS | 0.3367 | 0.3914 | 0.3912 | 0.2274 | 0.1172 | 0.2022 | 0.2029 | -0.0530 | 0.0801 | 0.1265 | 0.1134 | 0.0029 |
| w/ modification | 0.3362 | 0.3911 | 0.3910 | 0.2266 | 0.2116 | 0.2518 | 0.2523 | 0.1310 | 0.2185 | 0.2046 | 0.1891 | 0.2604 |
| GNN | 0.3500 | 0.3916 | 0.3924 | 0.2643 | 0.3177 | 0.3002 | 0.3042 | 0.3477 | 0.3197 | 0.2280 | 0.2176 | 0.5067 |

---

[7]The best shrinkage factor $\lambda$ for LLS modification is found to be $0.4$ for both CAMELS-SAM and CAMELS.

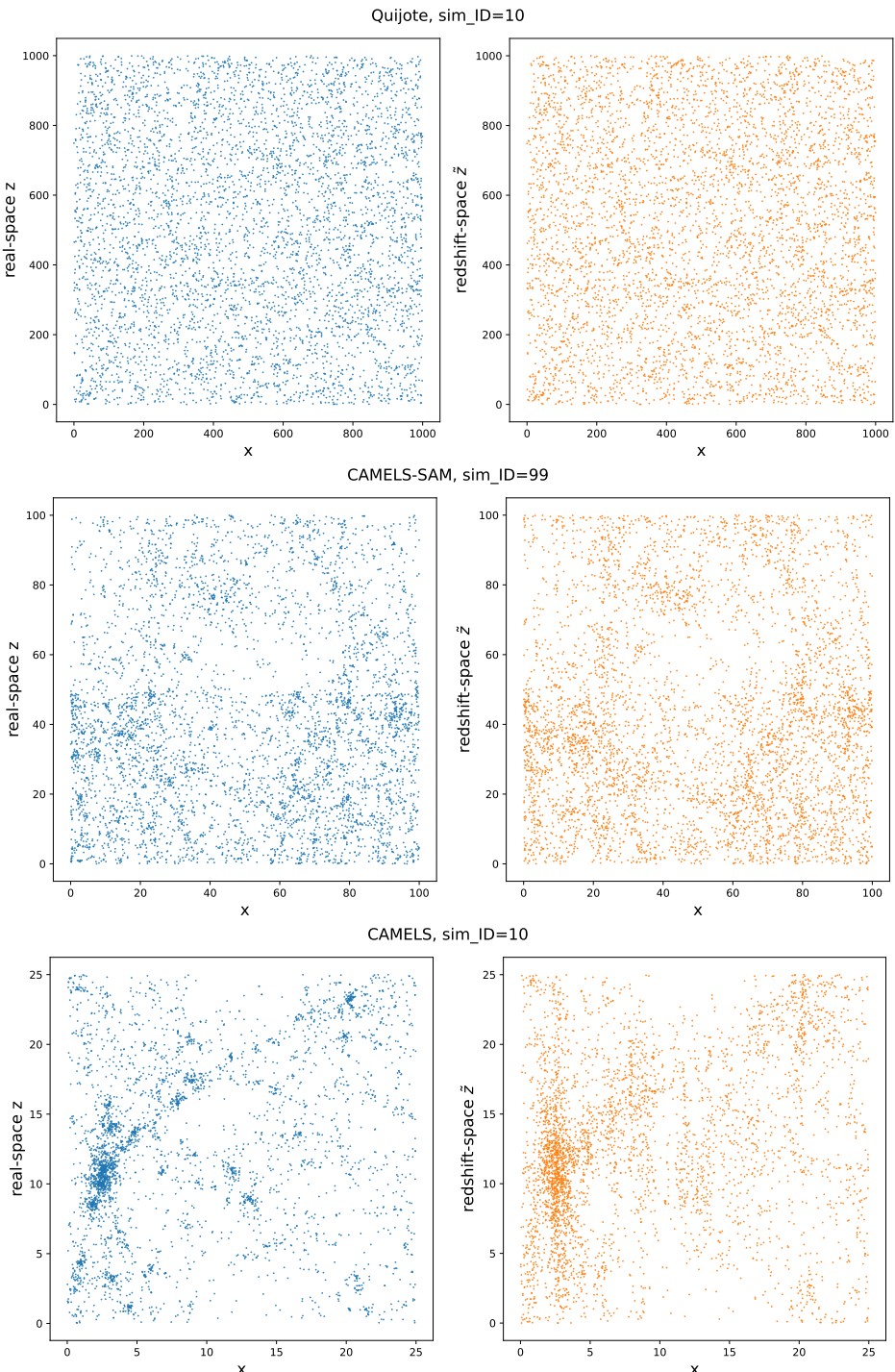

Figure B.4: Comparison between real-space positions (left panel) and redshift positions (right panel).

# C  Additional Merger Tree Experimental Results

## C.1  Predicting Cosmological Parameters from Merger Trees

**1-Nearest Neighbor predictor**  In this section we provide further details about the 1-Nearest Neighbor predictor introduced in Sec. 5.1. We consider for each tree the empirical distribution across nodes of three features: the concentration $c$, the maximum velocity $v_{\max}$, and the scale factor $a$ (a measure of time). Consider two empirical distributions, $\widehat{P}_A$ and $\widehat{P}_B$ of a univariate random variable $X$. The Kolmogorov-Smirnov statistic [e.g. 69] is

$$T_X = \max_x |\widehat{P}_A(X \leq x) - \widehat{P}_B(X \leq x)|. \tag{14}$$

This statistic only compares univariate distributions and there is no canonical extension to the multivariate case. In this paper, we define a straightforward discrepancy:

$$T_{(c,v_{\max},a)} = \sqrt{T_c^2 + T_{v_{\max}}^2 + T_a^2}. \tag{15}$$

While this statistics accounts for all variables, it ignores multivariate features, such as correlation between variables. For example, two distributions with the same marginals will satisfy $T = 0$, even though the multivariate distributions may be different, and so, strictly speaking, $T$ is not a distance.

There are many possible variations on this approach, which we leave to future work. First, we can consider different statistics or distances to compare merger trees, as discussed further in the next paragraph. Secondly, we can consider more features, including other variables such as the mass of each halo or the structure of the merger tree. We can also extend the model to $k > 1$ nearest neighbors. Finally, we can consider probabilistic models on trees, by viewing each tree as a collection of independent node partitions and leveraging suitable distributions such as the continuous categorical distribution proposed in [86].

**Distances for Trees**  We can view a graph neural network as a mapping from the input tree to a vector representation in the Euclidean space (such Euclidean representations are also known as *embeddings* in the graph learning community). From this perspective, another way to define distances for trees (e.g. merger trees) is to consider the Euclidean distances of their embeddings. Recently, Böker et al. [87] proved that the Euclidean distance of embeddings from message-passing neural network (MPNN) is topologically equivalent to the *tree distance*, a graph distance based on the fractional isomorphisms, as well as substructure counts via tree homomorphisms. A promising direction is to leverage these different distances defined on trees (or their embeddings) for predicting the cosmological parameters.

**Discussion of Results and Additional Visualizations**  As shown in Tab. 5 (left), $c$ and $v_{\max}$ are the most informative features when inferring $\Omega_m$, while $a$ is most informative for $\sigma_8$. Combining all nodes features achieves far superior accuracy. To examine the model performance further, we present scatter plots of the target cosmological parameters ($x$-axis) and the predicted parameters ($y$-axis) from (a) DeepSet and (b) GNN: Fig. C.1 shows the results from using the node feature scale factor $a$ only, whereas Fig. C.2 shows the results using all features (mass $M$, concentration $c$, halo maximum circular velocity $v_{\max}$, and scale factor $a$). We see mild improvements of GNN over DeepSet; both models are highly predictive of $\Omega_m$ when using all features. In this work, we make use of the entire merger tree from `CS-Trees` to predict cosmological parameters; using a subtree for such prediction task is an interesting direction for future research (e.g. the main branch used for generative modelling tasks in [88]).

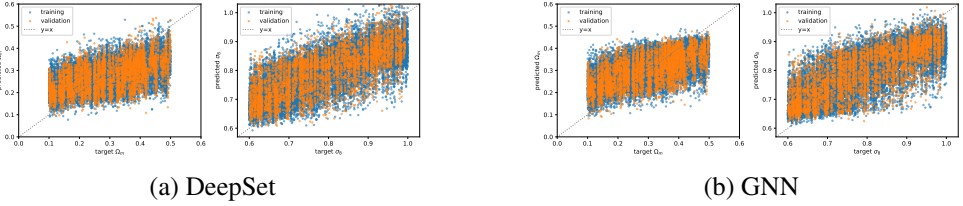

|        (a) DeepSet                    |        (b) GNN        |

Figure C.1: Cosmological parameter predictions using trees with node feature $a$.

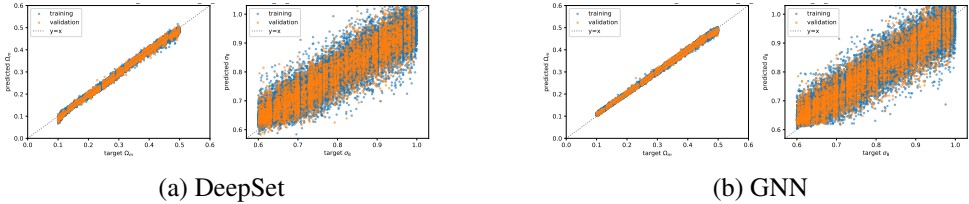

| (a) DeepSet | (b) GNN |

Figure C.2: Cosmological parameter predictions using trees with node features $(M, c, v_{\max}, a)$.

## C.2 Reconstructing Fine-Scale Merger Trees

**EPS Details** We adopt and extend the EPS implementation from [73][8]. The original algorithm is intended to generate many merger trees in parallel for a given set of constraints $s = \{\Omega_{\mathrm{m}}, \sigma_8, \ldots\}$, which include cosmological parameters and halo conditions (e.g., the range of root halo mass, the minimum mass threshold of the leaf halos). For our application in classifying unresolved merger nodes, each virtual node $v \in \tilde{V}$ in the coarsened tree $\mathcal{T}_c$ (see Algorithm 1) defines a specific constraint set $s_v$, derived from its post-merger node and pre-merger nodes. We collect a list of constraints $\mathbf{s} = [s_{v(1)}, s_{v(2)}, \ldots]$ induced from $20\%$ of virtual nodes across $n_{\mathcal{T}} = 97$ coarsened trees, and then iterate through each constraint $s \in \mathbf{s}$ to generate 5 merger trees using the EPS algorithm. The sequential processing over all constraints, of total size $|\mathbf{s}| = O(n_{\mathcal{T}} |\tilde{V}|)$, is currently the main computational bottleneck. We note that this step could be significantly accelerated by parallelizing over constraints, which we leave for future work.

**GNN Details** For the current GNN baseline, we use small-size GNNs with directed message-passing layers, as we observe that larger GNNs with higher hidden dimensionality overfit the data (with only 120 training trees). However, such (directed) message-passing GNNs are designed for generic (directed) graphs. A promising direction is to propose or leverage GNNs designed for directed trees, to further exploit the sparse connectivity and the directionality of the merger trees and achieve better accuracy.

**Discussion of Results** As reported in Tab. 5 (right), the ML approaches perform equivalently or better than the cosmology baseline (EPS). Moreover, they require considerably less time. Nevertheless, all baselines considered still have room for improvement. As a first step, we only consider predicting the unresolved merger node labels (i.e. whether there exists a merger node or not). Another important future step is to also predict the unresolved merger node features, which would enable enhanced inference quality from incomplete or coarse-grained merger trees.

---

[8]https://github.com/shergreen/SatGen

