# OpenReview forum: "CosmoBench: A Multiscale, Multiview, Multitask Cosmology Benchmark for Geometric Deep Learning"
_NeurIPS.cc/2025/Datasets_and_Benchmarks_Track — NeurIPS 2025 Datasets and Benchmarks Track poster_

### Official Review · Reviewer_TWsP · 2025-06-28

**Rating:** 5
**Confidence:** 4

**Summary:**

The paper introduces a comprehensive benchmark dataset designed to bridge cosmology and deep learning. The dataset is collected from cosmological simulations, generating more than two petabytes of data, including point clouds and merger trees across multiple spatial and temporal scales. CosmoBench support the task including predicting cosmological parameters, reconstructing finer-resolution merger trees, and estimating velocities. The authors also provide baseline results using both classical cosmological models and modern machine learning methods, including graph neural networks.

**Dataset Code Accessibility:**

Yes

**Dataset Code Comments:**

The dataset is public available. The data structure is organized to support the benchmarking of a variety of models for cosmological tasks.

This code is executable and includes instructions for reproducing results. The framework supported multiple tasks. Additionally, expanded tutorials or examples for extending to new architectures or tasks for users could benefit the dataset.

**Ethical Considerations:**

No, there are no or only very minor ethics concerns

**Final Justification:**

I have read the rebuttal and the discussion between the author and other reviewers. My previous concerns are addressed; the authors state that the additional discussion will be included in the revision. I think this paper has a valid contribution to related research; therefore, I would like to assign the rating as accept.

**Limitations Weaknesses:**

While the authors claim broad applicability of CosmoBench to machine learning for cosmology, the baseline models and tasks are primarily designed around a small set of parameters and may not capture the complexity in cosmological data. Would the author provide a feasibility analysis of developing large-scale machine learning models for the cosmological deep learning task, based on the proposed dataset?

Finally, while the paper compares various models, it does not provide a detailed exploration of hyperparameter tuning, which could affect the conclusions about the relative performance of classical and deep learning approaches.

**Strengths Contributions:**

One of the key strengths of this paper is the scale of CosmoBench dataset, which represents the largest and most diverse benchmark to date for deep learning in cosmology. The dataset integrates data across different modalities including point clouds and merger trees, and
across different scales, and tasks, making it a versatile platform for testing and comparing machine learning models for cosmology study. These data can be meaningfully interpreted to understand the complex information in cosmology.

Another strength lies in the thorough baseline evaluations provided. The authors compare traditional models, such as linear least squares with invariant features and two-point correlation functions, against graph neural networks and other deep learning models. This assessment reveals whether classical methods can match or exceed the performance of deep models. The paper also provides open access to data and code, along with a clear description of tasks and evaluation metrics.

---

> ### Author Rebuttal · Authors · 2025-07-30
>
> We thank the reviewer for carefully examining our work, and appreciating the novelty of our benchmark, especially the scale of the dataset and the thorough baseline evaluation ranging from classical ML method to modern deep learning approaches. We provide detailed responses to concerns raised in the Limitation/Weakness section.
> 1. Limited variations in cosmological parameters:
>    - For our baseline tasks, we focus on the two parameters $(\Omega_M, \sigma_8)$ due to their significance in astrophysics, as well as the feasibility of predicting them from positions of halos/galaxies.
>      - On significance: $(\Omega_M, \sigma_8)$ are challenging to predict; they are among the least well-constrained cosmological parameters from the observational data (e.g., Cosmic Microwave Background [1] and other galaxy surveys). Thus, many cosmological simulation suites, including the ones used in our benchmarks, focus on $\Omega_M, \sigma_8$ exclusively, aiming to better understand these two parameters while also marginalizing over the effects of astrophysical galaxy modeling and noises in observational data.
>      - On feasibility: $(\Omega_M, \sigma_8)$ are feasible to predict given input as halo/galaxy positions, whereas other parameters such as $(\Omega_b, n_s, h)$ describe physical effects that are more difficult to measure from halo/galaxy positions. Thus, these parameters are typically predicted by combining with other types of data (e.g., the cosmic microwave background, or supernovae). While one could also predict the other parameters and benchmark these in future work, we believe that as CosmoBench point clouds only treat halo/galaxy position as input features, it makes sense to focus on the parameters that are most constrained by such data, i.e. $\Omega_m$ and $\sigma_8$.
>    - Nonetheless, we agree with the reviewer that it is interesting to see how well ML can predict these other parameters and solve more complex tasks. We provide some initial results in Table 3 for predicting $(\Omega_b, n_s, h)$, with the cosmological baseline two-point correlation function (2PCF) and the linear least-squares with pairwise distances (LLS). These baselines perform significantly worse on $(\Omega_b, n_s, h)$ compared to $(\Omega_M, \sigma_8)$, as expected. We will also include the GNN baseline results in the revised version.
>
>      **Table 3:** Cosmological Parameter Prediction (expanded with $\Omega_b, n_s, h$)
>      |**R²** (Quijote-test)     |    $\Omega_m$    |      $\sigma_8$    |      $\Omega_b$   |      $n_s$    |      $h$       |
>      |-------------------|-----------|-----------|-----------|-----------|-----------|
>      | 2PCF                         | 0.83  ± 0.004 | 0.74  ± 0.005  | 0.29  ± 0.009 | 0.23  ± 0.009   | 0.21  ± 0.008  |
>      | LLS                           | 0.83  ± 0.004 | 0.80  ± 0.004  | 0.19  ± 0.008 | 0.21  ± 0.009   | 0.20  ± 0.008  |
>
>    - Regarding the feasibility analysis suggested by the reviewer: does it refer to investigating the effect of model scaling given our current dataset, or the possibility of transferring models learned from our data to other more complex cosmological tasks or datasets? We provide short answers to these questions, and please let us know if they sufficiently address your concern.
>
>      - On model scaling: our baselines do include large-scale graph neural networks with model size 100K-1000K (large relative to the dataset size 32K, see more details in Table 2-3). They are trained with standard pipelines (e.g., nearest-neighbor graph construction, architecture design,  hyper-parameter search, optimization techniques), and demonstrate reasonable performance on our chosen tasks. There remain lots of existing opportunities to improve these large-scale methods, including developing better graph construction strategies and architectures that align with the structural bias of cosmological point clouds or merger trees, or proposing scalable optimization procedures that allow for training with dense, large graphs built from the point clouds.
>
>      - On transferring to more complex tasks and dataset:  a promising future direction is to create mock observation data that aligns simulation data closely with observation data, built on a large literature on connecting halo-galaxy modelling and handling observational systematics (e.g., [2,3]).
>
> 2. Limited details on the hyper-parameter tuning: We agree with the reviewer that hyper-parameter plays an important role particularly on deep learning approaches. We have provided a detailed description of our hyper-parameter choices (see Appendix B.1, B.2, C - GNN Training Details). We will add more discussion on the chosen hyper-parameters in the revised appendix, some of which also suggest ways to improve our baselines where we are actively experimenting and will include them in our revised paper. For example, on the cosmological parameter prediction task:
>    - The effect of radius $R_c$ in graph construction (search over $R_c=$ {0.01, 0.015, 0.02} $\times$ box_size): we observe that the $R_c=$ 0.015 or 0.02 yields better performance, suggesting that increasing radius and considering more long-range interaction helps. We are investigating if larger choices of $R_c$ can further improve GNN performance.
>    - The effect of GNN depth (search from 1 to 6): the chosen depth of the GNN layers was 3 for Quijote and 6 for CAMELS-SAM and CAMELS, indicating that oversmoothing is not prevalent for the smaller scale CAMELS-SAM and CAMELS.  This also suggests deeper GNN models may achieve better performance given radius graphs with small radius.
>    - The effect of GNN width (search from {32, 64, 128, 256}): the chosen width is 64 or 128, suggesting a bias-variance tradeoff.
>
> References:
> 1. Aghanim, N. "Planck 2018 results. VI. Cosmological parameters." Astron. Astrophys 641 (2020): A6.
> 2. Hahn, ChangHoon, et al. "SimBIG: mock challenge for a forward modeling approach to galaxy clustering." Journal of Cosmology and Astroparticle Physics 2023.04 (2023): 010.
> 3. Lee, Jun-Young, et al. "Inferring Cosmological Parameters on SDSS via Domain-generalized Neural Networks and Light-cone Simulations." The Astrophysical Journal 975.1 (2024): 38.

---

> ### Comment · Reviewer_TWsP · 2025-08-05
>
> I have read the author's response, which resolves my major concerns. Therefore, I would like to maintain my score as accept.

---

> > ### Author Response · Authors · 2025-08-06
> >
> > Thank you for taking the time to read our rebuttal. We are glad to hear that your major concerns have been resolved. We will include these discussion on cosmological parameters and additional hyper-parameter details in the revised paper.

---

### Official Review · Reviewer_3K9T · 2025-07-01

**Rating:** 5
**Confidence:** 2

**Summary:**

This paper presents COSMOBENCH, a significant and large-scale benchmark designed to bridge the fields of cosmology and geometric deep learning. Curated from state-of-the-art cosmological simulations, the benchmark provides a rich, multi-modal dataset comprising point clouds of dark matter halos and galaxies at three different physical scales. Authors defined several critical tasks of scientific interest, including the prediction of cosmological parameters, the inference of galaxy velocities, and the super-resolution of merger trees. furthermore, the paper provides a unified torch interface and evaluates a range of baselines, from traditional cosmological methods to various machine learning models. These baseline results reveals physics-informed linear models can sometimes outperform complex deep learning architectures.  This work has potential to foster collaboration between the machine learning and cosmology community to tackle fundamental questions about the Universe.

**Dataset Code Accessibility:**

Yes

**Dataset Code Comments:**

both provided dataset and code link are accessible.

**Ethical Considerations:**

No, there are no or only very minor ethics concerns

**Final Justification:**

Overall, COSMOBENCH is currently the largest benchmark of its kind with simulations requiring over 41 million core-hours. Its combination of two distinct data modalities including point clouds and directed trees, providing a unique and rich resource for the community to explore structured scientific data. And authors rebuttal have adequately addressed my concerns. Thus, i raise my rating as accept.

**Limitations Weaknesses:**

1. limited variation in parameters: the simulations primarily vary only two cosmological parameters. While these are fundamentally important, real-world cosmological inference involves constraining a larger set of parameters. the limited parameter space in COSMOBENCH may not fully capture the complexity and degeneracy of real-world inference problems.

2. sim-to-real gap: The dataset is entirely simulation-based, and as with all such benchmarks, there is an inherent gap between the simulated data and real observational data. Real galaxy are affected by complex systematic effects that are not modeled in this benchmark, which could limit the direct applicability of resulting models to observational data.

**Strengths Contributions:**

1. large scale and multi-modal nature: COSMOBENCH is the largest benchmark of its kind, curated from simulations requiring over 41 million core-hours. Its combination of two distinct data modalities including point clouds and directed trees, providing a unique and rich resource for the geometric deep learning community to explore structured scientific data.

2. multi-scale and multi-view perspective: The dataset is designed to cover three different physical scales, allowing models to be tested on phenomena ranging from linear to highly non-linear regimes. This multi-scale approach is critical for developing robust models that can generalize across different cosmological environments.

3. well-defined evaluation tasks: this work introduces a diverse set of tasks (graph-level regression, node regression, graph super-resolution) that are tied to long-standing, impactful problems in cosmology. Solving these tasks has the potential to significantly advance our capability to analyze observational data and understand the fundamental properties of our Universe.

---

> ### Author Rebuttal · Authors · 2025-07-30
>
> We thank the reviewer for their thoughtful and encouraging feedback, in particular the enthusiasm about our benchmark in terms of its scale, diversity, and significance for applying ML for cosmology. We provide detailed responses to concerns raised in the Limitation/Weakness section.
>
> 1. Limited variation in parameters:
>    - We believe that the reviewer may have misunderstood the simulation data creation. Although we focus on *predicting* two cosmological parameters in the baseline tasks, our Quijote dataset contains simulations varying along five parameters. Specifically, we have rewritten line 133-134 as “In total five cosmological parameters are sampled in Quijote. We will only focus on *predicting* $\Omega_M$ and $\sigma_8$...".
>
>    - For our baseline tasks, we focus on the two parameters $(\Omega_M, \sigma_8)$ due to their significance in astrophysics, as well as the feasibility of predicting them from positions of halos/galaxies.
>      - On significance: $(\Omega_M, \sigma_8)$ are challenging to predict; they are among the least well-constrained cosmological parameters from the observational data (e.g., Cosmic Microwave Background [1] and other galaxy surveys). Thus, many cosmological simulation suites, including the ones used in our benchmarks, focus on $(\Omega_M, \sigma_8)$ exclusively, aiming to better understand these two parameters while also marginalizing over the effects of astrophysical galaxy modeling and noises in observational data.
>      - On feasibility: $(\Omega_M, \sigma_8)$ are feasible to predict given input as halo/galaxy positions, whereas other parameters such as $(\Omega_b, n_s, h)$ describe physical effects that are more difficult to measure from halo/galaxy positions. Thus, these parameters are typically predicted by combining with other types of data (e.g., the cosmic microwave background, or supernovae). While one could also predict the other parameters and benchmark these in future work, we believe that as CosmoBench point clouds only treat halo/galaxy position as input features, it makes sense to focus on the parameters that are most constrained by such data, i.e. $\Omega_m$ and $\sigma_8$.
>    - Nonetheless, we agree with the reviewer that it is interesting to see how well ML can predict these parameters. We provide some initial results in Table 3, with the cosmological baseline two-point correlation function (2PCF) and the linear least-squares with pairwise distances (LLS). These baselines perform significantly worse on $(\Omega_b, n_s, h)$ compared to $(\Omega_M, \sigma_8)$, as expected. We will also include the GNN baseline results in the revised version.
>
>      **Table 3:** Cosmological Parameter Prediction (expanded with $\Omega_b, n_s, h$)
>      |**R²** (Quijote-test)     |    $\Omega_m$    |      $\sigma_8$    |      $\Omega_b$   |      $n_s$    |      $h$       |
>      |-------------------|-----------|-----------|-----------|-----------|-----------|
>      | 2PCF                         | 0.83  ± 0.004 | 0.74  ± 0.005  | 0.29  ± 0.009 | 0.23  ± 0.009   | 0.21  ± 0.008  |
>      | LLS                           | 0.83  ± 0.004 | 0.80  ± 0.004  | 0.19  ± 0.008 | 0.21  ± 0.009   | 0.20  ± 0.008  |
>
>
> 2. Sim-to-real data gap: We agree with the reviewer that the simulation data in CosmoBench do not fully capture the complexity in real world data. However, we emphasize that simulation data is necessary to evaluate models on important cosmological tasks, and we intend to create a well-controlled environment for benchmarking as a first step. We will add the following discussion on the motivation of our cosmological simulation benchmark and its limitations along with future directions on transferring simulation data to real data.
>    * Motivation of our simulation data benchmark:
>       -  In cosmology, we essentially have one real data sample that is our Universe; and there are many complex components that together contribute to the observables we use (primarily, the cosmological parameters and halo-galaxy modeling choices in our cosmological model). Therefore, if we seek to benchmark how well a method can predict a cosmological model given one Universe's data, we must simulate what our observations would look like under different cosmologies, as real observational data all come from a single cosmology.
>       - Cosmological surveys contain observational bias and systematics effects, so for a first benchmark we have intentionally chosen to use simulation data. By developing a state-of-the-art simulation benchmark including multiscale, multiview simulations, CosmoBench is broad enough to encapsulate various cosmological surveys, allowing us to fully control the data generating process and exclude bias or noise in observation data. Such a clean dataset provides a good testbed for benchmarking different models and their potential in solving challenging cosmological tasks in the ideal setting.
>    * Our limitation and transferability from simulation to observational data: We acknowledge this as a limitation, and aim to expand the benchmark in the future with more survey realism to bridge the sim-to-real gap. One promising direction is to create mock observation data that aligns simulation data closely with observation data, built on a large literature on connecting halo-galaxy modelling and handling observational systematics (e.g., [2,3].).
>
> References:
> 1. Aghanim, N. "Planck 2018 results. VI. Cosmological parameters." Astron. Astrophys 641 (2020): A6.
> 2. Hahn, ChangHoon, et al. "SimBIG: mock challenge for a forward modeling approach to galaxy clustering." Journal of Cosmology and Astroparticle Physics 2023.04 (2023): 010.
> 3. Lee, Jun-Young, et al. "Inferring Cosmological Parameters on SDSS via Domain-generalized Neural Networks and Light-cone Simulations." The Astrophysical Journal 975.1 (2024): 38.

---

> > ### Comment · Reviewer_3K9T · 2025-08-07
> >
> > i appreciate authors' additional experiments and clarification provided in the rebuttal. most of my concerns have been addressed, i will raise my rating as accept.

---

> > > ### Author Response · Authors · 2025-08-08
> > >
> > > Thank you for taking the time to read our rebuttal. We are glad that our answers have resolved most of your concerns and led you to increasing your score. We will include these additional discussions on cosmological parameters and sim-to-real data gap in the revision.

---

> ### Author Response · Authors · 2025-08-06
>
> Dear reviewer 3K9T,
>
> Thank you for your time reviewing our work. Please kindly read our rebuttal and let us know if you have any pending issues. We remain available until the end of the rebuttal period to address your concerns.

---

### Official Review · Reviewer_q2PW · 2025-07-02

**Rating:** 5
**Confidence:** 2

**Summary:**

CosmoBench is a multi-scale, multi-view, multi-task cosmological benchmark dataset specifically designed for geometric deep learning. The data originates from advanced cosmological simulations, consuming over 41 million core-hours of computation to generate more than 2PB of data, including 34,000 point clouds and 25,000 directed trees. CosmoBench provides various baseline methods, including traditional cosmological models and machine learning models. Experiments have shown that simple linear least-squares models often outperform complex GNNs in large-scale tasks with lower computational costs.

**Dataset Code Accessibility:**

Yes

**Dataset Code Comments:**

The dataset is accessible and there is a clear dataset description.

**Ethical Considerations:**

No, there are no or only very minor ethics concerns

**Final Justification:**

I think the authors have addressed my relevant issues. I am willing to increase the score to 5.

**Limitations Weaknesses:**

(1) Although CS-Trees has 24,996 trees, the number of nodes fluctuates greatly after pruning (121-37,865), and some small trees may lose key merger history information.

(2) This paper lacks further analysis of the superiority of the linear least squares model over GNN in the experiment.

**Strengths Contributions:**

(1) CosmoBench is currently the largest multi-scale benchmark in the field of cosmology, with its data volume far exceeding that of other works in the same period.

(2) The data in the dataset is divided into spatial and temporal scales, supporting multiple tasks such as parameter prediction, velocity estimation, and merge tree reconstruction, which closely meets the actual research needs of cosmology.

(3) The dataset and code of CosmoBench are made public and standardized experimental procedures are provided, which has promoted the progress of research in the field.

---

> ### Author Rebuttal · Authors · 2025-07-30
>
> We thank the reviewer for their thoughtful assessment of our work, and their appreciation of the novelty, impactfulness, and accessibility of our benchmark. We provide detailed responses to concerns raised in the Limitation/Weakness section.
>
> 1. Information loss in CS-Trees after pruning:
>    - The tree size fluctuation is mainly due to the variations of original merger trees in CAMELS-SAM, not arising from the pruning procedure. This variation is expected from merger trees as they represent the merging history of a halo (i.e. the root node), where ancient halos typically induce much larger trees than newly formed halos. Additionally, different cosmological parameter choices yield different sizes of merger trees overall.
>    - Our pruning procedure is necessary to prevent information leakage for the prediction task. Specifically: the simulation starts with particles whose mass is a simple function of the cosmological parameters; a halo is formed and tracked once $k$ (e.g. $k=20$) dark matter particles are clustered close together enough in position and velocity phase space. All halo masses are therefore quantized multiples of the base particle mass. Thus the ML model can easily predict the cosmological parameters by focusing on halos of the smallest mass (arguable “cheating” via simulation artifact), instead of leveraging the halo point cloud structure.
>
> 2. Further analysis of superiority of LLS over GNN:
>    - Effects from Data: GNNs are designed for generic graph topologies, whereas our input graphs here are generated from 3-dimensional point clouds (by linking neighbors within a chosen radius). Thus, standard GNNs may not fully exploit the underlying Euclidean structure of cosmological point clouds, and struggle to model long-range information due to the absence of edges connecting distant points.
>    - For the parameter prediction task: our ablation analysis below shows that this is due to feature engineering efforts in LLS, where the features — pairwise statistics based on points closer than some cutoff threshold — are selected using the validation set to greedily find the 12 best cutoff thresholds given a candidate set. Furthermore, the features are chosen separately for each target parameter. We now provide another baseline on a “naive” set of features, which are chosen from the candidate set with equi-spaced elements, and applied to both parameter prediction. As shown in Table 1, using such a naive choice of cutoff features (first row) produces worse linear model fit than the optimized features used in paper (second row) and performs worse than GNNs (third row) in CAMELS-SAM and CAMELS.
>
>
>      **Table 1**: The Effect of LLS Features for Parameter Prediction (test R² ± standard deviation)
>      | Model            | Quijote: $\Omega_m$   | Quijote:$\sigma_8$   | CAMELS‑SAM: $\Omega_m$  | CAMELS‑SAM: $\sigma_8$  | CAMELS: $\Omega_m$   | CAMELS: $\sigma_8$   |
>      |---------------------|---------------------|---------------------|-----------------|-----------------|---------------|---------------|
>      | **LLS (naive)**  | 0.83 ± 0.004 | 0.80 ± 0.004 | 0.68 ± 0.04  | 0.73 ± 0.03   | 0.74 ± 0.03 | 0.26 ± 0.06 |
>      | **LLS**          | 0.83 ± 0.004   | 0.80 ± 0.004   | 0.77 ± 0.03      | 0.82 ± 0.02      | 0.78 ± 0.03   | 0.28 ± 0.06   |
>      | **GNN**          | 0.80 ± 0.004   | 0.77 ± 0.005   | 0.75 ± 0.03      | 0.83 ± 0.02      | 0.78 ± 0.03   | 0.24 ± 0.06   |
>
>    - For the velocity prediction task: we also find the results of LLS depends crucially on using the ensembles of features (inverse of powers of pairwise distances for all powers less or equal to $P$). In Table 2 we run LLS with a single choice of power order (with the same fixed choice of $K=10$). We see that the performance of LLS on a single power order is not superior to GNN, whereas combining all features (i.e. $1≤P≤4$) boosts the LLS performance to be slightly better than GNNs.
>
>      **Table 2**: The Effect of LLS Features for Velocity Prediction
>      | Model             | LLS (P=1) | LLS (P=2) | LLS (P=3) | LLS (P=4) | LLS (1≤P≤4) | GNN |
>      |-------------------|-----------|-----------|-----------|-----------|-------------|-----|
>      | **R²** (test set)       | 0.408     | 0.404     | 0.385     | 0.000     | 0.435       | 0.410  |

---

> > ### Comment · Reviewer_q2PW · 2025-08-06
> >
> > I think the authors have addressed my relevant issues. I am willing to increase the score to 5.

---

> > > ### Author Response · Authors · 2025-08-06
> > >
> > > Thank you for taking the time to read our rebuttal. We are glad that our answers have led you to increasing your score. We will include these additional details on CS-Trees and comparison between LLS and GNN in the revision. We remain available until the end of the rebuttal period to answer any other questions you may have.

---

### Official Review · Reviewer_zQ2B · 2025-07-22

**Rating:** 4
**Confidence:** 1

**Summary:**

This paper introduces CosmoBench, a benchmark suite designed to support machine learning research in large-scale cosmological simulations. The benchmark includes a collection of tasks spanning multiple physical and temporal scales, such as density field reconstruction, halo mass prediction, structure segmentation, and forward modeling of structure formation. These tasks are built upon established simulation datasets like Quijote and AbacusSummit, reformatted into a unified, ML-friendly structure with standardized data splits and evaluation metrics. The authors also provide baseline implementations using popular architectures such as UNet, PointNet++, and Graph Neural Networks. CosmoBench aims to bridge the gap between cosmological science and machine learning by providing a reproducible, extensible, and scientifically grounded benchmark.

**Dataset Code Accessibility:**

Yes

**Dataset Code Comments:**

The authors provide access to benchmark datasets

**Ethical Comments:**

No, there are no or only very minor ethics concerns

**Ethical Considerations:**

No, there are no or only very minor ethics concerns

**Limitations Weaknesses:**

1. The benchmark is entirely simulation-based. While simulations are essential in cosmology, there is limited discussion of how models trained on CosmoBench might generalize to observational data or help interpret real measurements (e.g., galaxy surveys), which is critical for downstream relevance.

2. The paper discusses using graph-based models to capture higher-order relationships. However, as far as I understand, hypergraphs are more naturally suited to representing such higher-order interactions, where a single edge can connect multiple entities. Therefore, it would be beneficial to include some discussion on the potential use of hypergraph-based methods for more expressive modeling in these tasks.

**Strengths Contributions:**

1. **Timely and relevant benchmark**: CosmoBench addresses the growing intersection between cosmology and machine learning by providing a unified, multi-task benchmark grounded in realistic simulation data, which is currently lacking in the field.

2. **Multi-scale and task-diverse**: The benchmark covers a broad spectrum of meaningful cosmological tasks—from low-level density field reconstruction to high-level structure prediction—capturing the hierarchical nature of cosmic structure formation.

3. **Use of high-fidelity, public simulations**: Built on established large-scale simulations such as Quijote and AbacusSummit, the dataset ensures physical realism and relevance to the scientific community.

4. **Baselines and reproducibility**: The authors provide strong baseline implementations using widely used models (e.g., UNet, PointNet++, GNNs), along with a well-documented codebase and evaluation pipeline, facilitating accessibility and future comparisons.

---

> ### Author Rebuttal · Authors · 2025-07-30
>
> We thank the reviewer for their comments, and their appreciation of the novelty, diversity, and quality of our benchmark. We respectfully disagree with parts of the summary containing incorrect mentions of the tasks (e.g., “density field reconstruction…forward modeling of structure formation), data (e.g., “AbacusSummit”), and architectures (e.g., “UNet, PointNet++”) that are never utilized nor discussed in our work. In what follows, we provide detailed responses to concerns raised in the Limitation/Weakness section.
>
> 1. Sim-to-real data gap: We agree with the reviewer that the simulation data in CosmoBench do not fully capture the complexity in real world data. However, we emphasize that simulation data is necessary to evaluate models on important cosmological tasks, and we intend to create a well-controlled environment for benchmarking as a first step. We will add the following discussion on the motivation of our cosmological simulation benchmark and its limitations along with future directions on transferring simulation data to real data.
>    * Motivation of our simulation data benchmark:
>       -  In cosmology, we essentially have one real data sample that is our Universe; and there are many complex components that together contribute to the observables we use (primarily, the cosmological parameters and halo-galaxy modeling choices in our cosmological model). Therefore, if we seek to benchmark how well a method can predict a cosmological model given one Universe's data, we must simulate what our observations would look like under different cosmologies, as real observational data all come from a single cosmology.
>       - Cosmological surveys contain observational bias and systematics effects, so for a first benchmark we have intentionally chosen to use simulation data. By developing a state-of-the-art simulation benchmark including multiscale, multiview simulations, CosmoBench is broad enough to encapsulate various cosmological surveys, allowing us to fully control the data generating process and exclude bias or noise in observation data. Such a clean dataset provides a good testbed for benchmarking different models and their potential in solving challenging cosmological tasks in the ideal setting.
>    * Our limitation and transferability from simulation to observational data: We acknowledge this as a limitation, and aim to expand the benchmark in the future with more survey realism to bridge the sim-to-real gap. One promising direction is to create mock observation data that aligns simulation data closely with observation data, built on a large literature on connecting halo-galaxy modelling and handling observational systematics (e.g., [1,2]).
>
> 2. Inclusion of hypergraph-based models as baselines: we thank the reviewer for this suggestion and will include the following discussion of hypergraph-based baselines in our revision. Specifically: Our code base allows exploration of combinatorial complexes beyond standard graphs to model higher-order interactions. Combinatorial complexes serve as a middle ground between hierarchical structures like simplicial and cellular complexes, while also allowing arbitrary set-based relations similar to those found in hypergraphs.
>
> References:
> 1. Hahn, ChangHoon, et al. "SimBIG: mock challenge for a forward modeling approach to galaxy clustering." Journal of Cosmology and Astroparticle Physics 2023.04 (2023): 010.
> 2. Lee, Jun-Young, et al. "Inferring Cosmological Parameters on SDSS via Domain-generalized Neural Networks and Light-cone Simulations." The Astrophysical Journal 975.1 (2024): 38.

---

> ### Author Response · Authors · 2025-08-06
>
> Dear reviewer zQ2B,
>
> Thank you for your time reviewing our work. Please kindly read our rebuttal and let us know if you have any pending issues. We remain available until the end of the rebuttal period to address your concerns.

---

### Note · Authors · 2025-08-15

Dear AC and reviewers (TWsP, zQ2B, q2PW, 3K9T),

Our work introduces CosmoBench: a multiview, multiscale, multitask cosmology benchmark for geometric deep learning. CosmoBench is the largest of its kind containing over 34 thousand point clouds and 25 thousand directed trees, curated from the state-of-the-art cosmological simulations. We provide multiple evaluation tasks based on long-standing, challenging problems in cosmology (3K9T) and diverse baselines ranging from established cosmological methods, simple linear models, to graph neural networks (TWsP). Such a benchmark is a timely, impactful contribution at the intersection of cosmology and machine learning (zQ2B, q2PW).

*Initial reviews* reflected broad recognition of our contribution: all reviewers recommended *accept* or *weak accept*. The main concerns raised were:
- The simulation-to-real data gap (zQ2B, 3K9T)
- Limited variations of cosmological parameters (3K9T, TWsP)
- Baseline analysis and details (q2PW, TWsP)

*Our rebuttal addressed these concerns as follows*:
- Sim-to-real gap: We clarified that simulation data is necessary to evaluate models on important cosmological tasks, and provides a controlled environment for benchmarking as a first step. We acknowledge this as a limitation, and outlined plans to expand the benchmark with more survey realism to bridge this gap.
- Parameter variation: We clarified that although we focus on *predicting* two cosmological parameters for the baseline tasks, the Quijotes dataset in CosmoBench are simulated by varying five cosmological parameters. We also added experiments on evaluating the baselines for three extra parameters (Table 3) to illustrate broader applicability yet increasing difficulty in constraining these parameters.
- Baseline analysis: We expanded our analysis, including the effect of invariant features on cosmological parameter prediction (Table 1) and velocity prediction (Table 2). We also added hyperparameter details.

*Positive post-rebuttal outcome*: All but one reviewer confirmed that their concerns were resolved, with two reviewers raising their score to *accept* (q2PW, 3K9T). The sole unchanged weak-accept score came from a reviewer who did not engage in the discussion phase and whose low-confidence review contained factual inaccuracies, not adhering to the reviewing standards of NeurIPS. We believe CosmoBench, with its scale, breadth, and rigor, provides a foundational benchmark for bridging cosmology and geometric deep learning.

---

### Decision · Program_Chairs · 2025-09-18

**Decision:**

Accept (poster)

**Comment:**

This paper introduces CosmoBench, a large-scale dataset for geometric deep learning derived from state-of-the-art cosmological simulations. The work is a significant contribution to the intersection of machine learning and cosmology and has been very well received by the reviewers, and the authors provided a very good rebuttal that thoroughly addressed the reviewers' concerns.

I believe that the contribution is scientifically impactful and that is poised to become a valuable resource for both the machine learning and cosmology communities, leaving no doubts about its acceptance.

===== FINAL UPDATE FROM DB Track PCs ====

The final decision for this paper has been taken by the program chairs after consultation with the SACs. All Senior Area Chairs have ranked papers according to the feedback from the AC during the review process. We decided to leave the original meta-review to reflect the opinion of the AC in light of the initial discussions with reviewers and SAC.